# Effect of local and remote sources and new particle formation events on the activation properties of cloud condensation nuclei in the megacity of São Paulo, Brazil

Carlos Eduardo Souto-Oliveira[1,2], Maria de Fátima Andrade[3], Prashant Kumar[4,5], Fábio Juliano da Silva Lopes[3,6], Marly Babinski[1], Eduardo Landulfo[6].

[1]Geocronological Research Centre, Institute of Geosciences, University of São Paulo, São Paulo, 05508-080, Brazil
[2]Chemistry Research Centre, UNIFIEO, São Paulo, 06018-903, Brazil
[3]Department of Atmospheric Sciences, Institute of Astronomy, Geophysics and Atmospheric Sciences, University of São Paulo, São Paulo, 05508-090, Brazil
[4]Department of Civil and Environmental Engineering, Faculty of Engineering and Physical Sciences (FEPS), University of Surrey, GuilfordGU2 7XH, United Kingdom
[5]Environmental Flow (EnFlo) Research Centre, Faculty of Engineering and Physical Sciences (FEPS), University of Surrey, GuildfordGU2 7XH, United Kingdom
[6]Nuclear and Energy Research Institute, IPEN-CNEN, Centre for Laser and Applications (CLA), São Paulo, 05508-970, Brazil

*Correspondence to*: Carlos Eduardo Souto-Oliveira (carlos.edu.oliveira@usp.br)

**Abstract.** Atmospheric aerosol is the primary source of cloud condensation nuclei (CCN). The microphysics and chemical composition of aerosols can affect cloud development and the precipitation process. Among studies conducted in Latin America, only a handful have reported the impact of urban aerosol on CCN activation parameters such as activation ratio (AR) and activation diameter ($D_{act}$). With over 20 million inhabitants, the Metropolitan Area of São Paulo (MASP) is the largest megacity in South America. To our knowledge, this is the first study to assess the impact that remote sources and new particle formation (NPF) events have on CCN activation properties in a South American megacity. The measurements were conducted in the MASP between August and September 2014. We measured the CCN within the 0.2–1.0% range of supersaturation, together with particle number concentration (PNC) and particle number distribution (PND), as well as trace-element concentrations and black carbon (BC). NPF events were identified on 35% of the sampling days. Combining trace-element and BC data with an aerosol profile from Lidar and HYSPLIT model analyses allowed us to identify the contribution of sea salt and biomass burning from remote regions on 28% and 21% of the sampling days, respectively. The AR and $D_{act}$ parameters showed distinct patterns for diurnal and nocturnal periods. For example, CCN activation was lower during the diurnal periods than during the nocturnal periods, a pattern that was found to

be associated mainly with local road traffic emissions. A decrease in CCN activation was observed on the NPF event days, mainly due to high concentrations of particles with smaller diameters. We also found that aerosols from sea salt and biomass burning had minor effects on $D_{act}$. For example, nights with biomass burning showed slightly higher $D_{act}$ values than did non-event nights. Our results show that particulate matter from local pollution sources, mainly local road traffic emissions, has a greater effect on CCN activation parameters than those from remote sources.

## 1 Introduction

Atmospheric aerosol has been the focus of numerous studies, because of its adverse effects on public health and climate (Heal et al., 2012). Aerosol particles are a significant source of cloud condensation nuclei (CCN). Owing to their microphysics and chemical composition, aerosol particles can affect cloud development and the precipitation process. The high concentrations of CCN in the air favour the formation of clouds with small droplets, which can lead to suppression of precipitation in shallow and short-lived clouds (Andreae and Rosenfeld, 2008). Such phenomena have been observed at different places. A study carried out in the Brazilian Amazon showed a reduction in precipitation during biomass burning events, due to greater scavenging from smoky clouds than from clouds formed under clean (blue and green ocean) air masses. The smoky clouds were shown to form at higher altitudes (> 4 km), due to their smaller droplet size, and to produce hydrometeors including isolated intense showers and thunderstorms (Andreae et al., 2004; Freud et al., 2008).

The CCN activation properties can be estimated by parameters such as activation ratio (AR) and activation diameter ($D_{act}$). In several studies representing different locations of the world, these parameters have been employed for determining the efficiency of aerosol particles in acting as CCN. The variability of these parameters is dependent on size and chemical composition of particles. In general, $D_{act}$ is more sensitive to the chemical composition of aerosols, whereas the AR is related to particle size and chemical composition. Large particles, inorganic salts and polar organic molecules are typically more hygroscopic than are smaller particles and non-polar molecules (e.g. hydrocarbons). Some studies have reported that aerosol from air masses containing pollutants (e.g. urban, fresh traffic and biomass burning emissions) are less efficient in activating CCN (>$D_{act}$ and <AR) than are non-polluted (e.g. forest and marine) air masses (Frank et al., 2006; Andreae and Rosenfeld, 2008; Quinn et al., 2008; Furutani et al., 2008; Bulkart et al., 2011; Deng et al., 2013; Leng et al., 2013; Paramonov et al., 2013). However, studies carried out at different locations and under different environmental conditions have demonstrated that aged polluted aerosols present CCN activation properties similar to those of non-polluted or even marine environment air (Peng et al., 2014). Those observations are supported by the growth of particles, atmospheric processing of non-soluble organics and mixture with more hygroscopic compounds such as sulphate, nitrate and organic acids (Andreae et al., 2004; McFiggans et al., 2006; Quinn et al., 2008; Furutani et al., 2008; Dusek et al., 2010; Bougiatioti et al., 2011; Zhang et al., 2014).

Aerosols originate from primary and secondary processes. Primary aerosols are emitted into the atmosphere directly by anthropogenic and natural sources. In contrast, secondary aerosols are formed in the atmosphere through the nucleation of no- or low-volatile gases containing inorganic and organic compounds, followed by growth to larger particles (Holmes, 2007). New particle formation (NPF) directly affects particle number concentration (PNC), particle size distribution (PND) and the optical properties of aerosols, as well as the properties of CCN and clouds. These events have been recently studied in urban, coastal and forest environments (Kulmala et al., 2004; Dal Maso et al., 2005; Backman et al., 2012; Betha et al., 2013; Peng et al., 2014; Nie et al., 2014; An et al., 2015; Sorribas et al., 2015; Kumar et al., 2014). However, there have been only a few studies evaluating the effect of NPF on CCN properties (Dusek et al., 2010; Yue et al., 2011; Sihto et al., 2011; Paramonov et al., 2013).

Typically, nucleation events occur in clean air under high solar radiation conditions. Many authors have shown that sulphuric acid and precursor species ($SO_2$, hydroxyl, $NH_3$ and oxidised organic compounds) play important roles in the nucleation process (Yue et al., 2011; Andreae, 2013; Long et al., 2016). Reche et al. (2011) suggested that the occurrence of $SO_2$ peaks contributes to midday nucleation bursts as a function of the sources. Kumar et al. (2014) discussed the different conditions for the secondary formation of particles over different types of urban areas. Recent studies have demonstrated the importance of oxidised organic vapors to drive NPF nucleation with $H_2SO_4$ and enhance secondary particle growth (Metzger et al., 2010; Donahue et al., 2013). Zhu et al. (2014) demonstrated the importance of SOA to particle growth over urban sites with different levels of pollution.

Volatile organic compounds (VOCs) constitute a fundamental precursor of secondary organic aerosols (SOA) and tropospheric ozone ($O_3$). Primary organic aerosols (POA) originate from biogenic sources (isoprene, terpenes, dimethylsulphide and dicarboxylic acids) and anthropogenic sources (biomass burning and traffic), thereafter being emitted directly into the atmosphere. Atmospheric species such as hydroxyl radicals and $O_3$ play a major role in VOC chemical degradation and the consequent formation of SOAs, which contain polar oxygenated functional groups (Hallquist et al., 2009). Recent studies have confirmed that the SOA yield is dependent on high concentrations of nitrogen oxides ($NO_x$), which explain the formation of certain SOAs, such as those derived from isoprene degradation (Shilling et al., 2013; Yuan et al., 2013). Another study, carried out in California, showed that vehicle emissions play an important role in the formation of urban SOAs (Ortega et al., 2016). In MASP, biofuels (ethanol and biodiesel) increase the emission of carbonyl compounds, which can be precursors of secondary oxygenated pollutants. In one study conducted in the MASP, Oyama et al. (2015) showed the emission factors for light-duty vehicles, which run on gasohol or ethanol, and for heavy-duty vehicles, which run on biodiesel. The authors found that oxygenated hydrocarbon compounds accounted for a major proportion of the aerosol composition. Those same authors also reported that, during biodiesel combustion, heavy-duty vehicles in the MASP emit greater quantities of volatile nitrogen compounds, which are associated with the $NO_x$ chemistry, than light-duty vehicles in the MASP. Wallington et al. (2016) showed that engine calibration is a determinant of $NO_x$ emissions, which are higher from biodiesel-burning vehicles. The use of biofuels has introduced new challenges for the description of atmospheric chemistry, by increasing the emissions of carbonyl and polycyclic aromatic hydrocarbons (PAHs, including those containing nitrogen

and those that are oxygenated), as shown by Karavalakis et al. (2011).

With over 20 million inhabitants, the MASP is the biggest megacity in South America. Therefore, the MASP represents an important city for global-scale atmospheric studies. The MASP fleet comprises more than 7 million vehicles, which constitute one of the main sources of particles, together with a variety of industries and construction activities (IBGE, 2014).

To date, there has been only one study measuring and modelling CCN in the MASP (Almeida et al., 2014). In general, there has been a limited number of studies reporting CCN, PNC and PND measurements, comparing Latin American cities with those in Europe and United States in terms of the effects that those properties have on the atmosphere (IPCC, 2013).

The aim of this study was to evaluate CCN activation properties under the influence of local and remote sources, as well as to identify NPF events. We measured CCN simultaneously with PNC, PND, black carbon (BC) and trace-element. PND

contour plots were constructed in order to identify and classify NPF events. Light detection and ranging (Lidar) analyses and Hybrid Single Particle Lagrangian Integrated Trajectory (HYSPLIT) analyses (Draxler and Rolph, 2003) were combined with trace-element and BC size distribution data in order to identify air masses from remote sources in the MASP. We evaluated the effects that those air masses associate with local sources and NPF events had on CCN activation properties. To our knowledge, this is the first study evaluating the effects of remote sources and NPF events on CCN activation properties

in a megacity in South America.

## 2 Methodology

### 2.1 Site description and meteorological data

The SPMA (23.5°S; 46.6°W) is an urban region encompassing an area of 7946 km$^2$, in the south-western region of Brazil (Fig. 1). With over 20 million inhabitants, the MASP is the largest megacity in South America and rank among the ten most

populous in the world. The MASP is situated in a sedimentary basin within the state of São Paulo (the Tietê River valley, which has a mean elevation of 720 m), approximately 45 km north-west of the Atlantic Ocean.

The measurements were taken on the rooftop of the *Instituto de Astronomia, Geofísica e Ciências Atmosféricas* (IAG, Institute of Astronomy, Geophysics and Atmospheric Sciences) on the main campus of the University of São Paulo (USP, 23.56°S; 46.73°W), which is situated in the western region of the city. There are two main roads (the Marginal Pinheiros and

the Marginal Tietê), approximately1 km northeast and 4 km north of the sampling site, respectively. During the diurnal period (0700–1900 h, local time), those roads have a high mean hourly traffic density (8000 automobiles, 1500 motorcycles, 220 trucks and 90 buses). At night, traffic volume is less than half of what is noted during the day and is dominated by heavy-duty vehicles (mainly trucks), which are not allowed to circulate during the daytime hours. Other important sources of particles in the MASP are construction sites on and near the USP campus and a cement industry about 2 km to the west. The

major industrial areas (Mauá and Cubatão) are respectively located approximately 30 and 45 km to the south-east, the industrial city of Cubatão being located in the coastal region (Fig. 1).

## 2.2 Measurements of CCN

The CCN were counted in a single-column continuous-flow stream-wise thermal gradient chamber (CCN-100; Droplet Measurement Technologies, Boulder, CO, USA). The supersaturated water vapor is generated inside the column centre line (radius, 0.9 mm; length, 360 mm; and wall thickness, 8 mm), which is controlled by the gradient temperature. When the supersaturation (SS) generated is greater than the critical SS, the particle grows to the critical $D_{act}$, after which it has the potential to become a cloud droplet. As the activated particles leave the column chamber, those in the 0.75–10 µm size range are counted by an optical particle counter, which employs (OPC) a standard light scattering technique (Roberts and Nenes, 2005).

All samples were collected between 19 August and 3 September 2014. The CCN counter was operated at a flow rate of 0.5 L $min^{-1}$ with variable supersaturation (0.2, 0.4, 0.6, 0.8 and 1.0%). Each cycle (~25 min) corresponds to measurements of all supersaturation, each supersaturation being measured during 5-min; our analysis considered only the data for steady-state supersaturation.

## 2.3 Measurements of size-resolved particles

The PND and PNC were measured with a scanning mobility particle sizer (SMPS, model3936; TSI Inc., St. Paul, MN, USA). The SMPS consists of a differential mobility analyser (DMA, model 3080; TSI Inc.) and a condensation particle counter (CPC, model 3010; TSI Inc.). The DMA separates particles according to their electrical mobility, and the particles that pass through the DMA column are counted by the CPC (Kumar et al., 2010). In the CPC, particles grow into droplets by condensation of alcohol vapor and are then counted by a laser beam.

The aerosol size distribution was measured in the 10–450 nm range, particles being scanned in 22 diameter size bins, with a 5-min time resolution. The gas sample and sheath flow rate were 1.0 and 6.0 L $min^{-1}$, respectively. Determining PNCs from the SMPS has been found to result in the undercounting of particles during ambient measurements, mainly due to lower DMA transfer probability or deviation in sampling and sheath flow rates (Almeida et al., 2014). Another deviation is related to the different diameter size range of particles measured by the SMPS (10–450 nm) and CCN (<10 µm), which can lead to overestimation of AR values, as calculated from the CCN/PNC ratio. A correction factor of 1.3 was applied to the entire data set in order to correct for undercounting during the measurement of PNCs and for overestimation of the AR. That factor was obtained by linear fitting of scatter plot data (CCN versus PNC) with AR values>1.

## 2.4 Measurements of BC and trace-element species

A Micro-orifice Uniform Deposit Impactor (MOUDI) was used in order to collect particles measuring <0.056, 0.05618, 0.1, 0.18, 0.32, 0.56, 1.0, 1.8, 3.2, 5.6 and 10 µm. The MOUDI selects particles by the size of its orifices and operates with a sampling air flow of 30 L $min^{-1}$. In each stage, the particles larger than the size of micro-orifice were collected on polycarbonate filters (Whatman, 47 mm diameter, 8 µm pore size). The sampling intervals were 12h (0700–1900 h and

1900–0700 hm local time). After the sampling, the filters were stored in a temperature- and humidity-controlled room for subsequent analysis.

The aerosol mass concentrations were determined gravimetrically before and after the sampling; the filters were weighed on a microbalance with a 1µg readability (Mettler-Toledo, Columbus, OH, USA). Elemental concentrations were determined by

Energy Dispersive X-ray Fluorescence Spectrometer (EDX 700; Shimadzu Corporation, Analytical Instruments Division, Tokyo, Japan). The spectrometer operates at 5–50 kV and 1–1,000 µA, using a low-power Rh-target tube, and the elemental characteristic X-ray radiation emitted from the aerosol sample is detected with an Si(Li) detector. The spectra obtained were processed and quantified with the WinQXAS program, available from the International Atomic Energy Agency (http://www.iaea.org/OurWork/ST/NA/NAAL/pci/ins/xrf/pciXRFdown.php).

We measured BC by optical reflectance with a smoke stain reflectometer (model 43D; Diffusion Systems Ltd, London, UK). Visible light was emitted from a tungsten lamp, and the reflected radiation was detected. The empirical calibration curve that converts the reflected signal to BC concentrations was obtained. The light absorption by the carbonaceous fraction present in atmospheric aerosol is correlated with elemental carbon.

### 2.5 Monitoring of gases and ultraviolet radiation

To determine the importance of the concentrations of certain gases ($SO_2$, $O_3$, NO and $NO_x$) and ultraviolet (UV) radiation on NPF event days, we considered the hourly values of the compounds measured at air quality stations operated by the São Paulo State *Companhia de Tecnologia de Saneamento Ambiental* (CETESB, Environmental Protection Agency). The $O_3$ and UV radiation were measured ata CETESB station on the USP campus, approximately 500m from the IAG. The $SO_2$, $O_3$, NO and $NO_x$ data were obtained from four CETESB stations located off of the USP campus: 7 km to the north-west (Osasco); 5

km to the north (Marginal Pinheiros); 10km to the south-east (Congonhas); and 5 km to the south-west (Taboão da Serra). Those four stations were chosen in order to represent the influence of wind direction on the behaviour of the particulate matter concentrations.

### 2.6 Lidar and trajectory analyses

A Raman lidar system installed at the Brazilian Nuclear and Energy Research Institute, designated *Lidar do Município de*

*São Paulo I* (MSP-Lidar I, lidar I of the Municipality of São Paulo), approximately 400 m from the IAG site, was employed in order to take independent measurements related to particle extinction profiles and backscatter coefficients, thus determining the particle extinction-to-backscatter ratio, at 532 and 355 nm, respectively. The MSP-lidar I system comprises a commercial Nd:YAG laser operating at 355 nm and 532 nm. The output energy per pulse is 400 mJ for 532 nm and 230 mJ for 355 nm, at a repetition rate of 10 Hz and pulse duration of 5 ns. The light beam is expanded by a factor of 5 in order to

reduce the divergence of the expanded beam to less than 0.5 mrad. The laser beam is vertically directed to the atmosphere and the backscattered radiation is collected with a coaxial Newtonian telescope that has a primary mirror diameter of 300 mm and a focal length of 1.5 m. The receiver field of view is set to 0.1mrad, providing a complete overlap between the laser

beam and the telescope field of view at altitudes of approximately 300 m above the lidar system. After the photons had been separated and had passed through the respective interference filters, we used photomultiplier tubes (R9880U-110; Hamamatsu Photonics K.K., Hamamatsu, Japan) in order to identify those backscattered elastically, detected at 355 nm and 532 nm, and those scattered inelastically by nitrogen molecules (Raman-shifted protons), detected at 387 nm and 607 nm.

The 355 nm and 532 nm filters have a bandwidth of 1 nm, whereas the 387 nm and 607 nm filters have a bandwidth of 0.25 nm. A transient digitiser operating in analogue and photon counting modes recorded data at 12-bit resolution. Data were averaged every 2 min, with a typical height resolution of 7.5 m. The MSP-Lidar I system, which belongs to the Latin American Lidar Network (http://lalinet.org), a federative coordinated lidar network focused on the vertically-resolved monitoring of the particle optical properties distribution over Latin America (Guerreo-Rascado et al., 2014; Guerreo-

Rascado et al., 2016), has been at the USP station since 2008.

Elastic backscatter lidar data and air mass modelling trajectories were used in order to evaluate the relationship that trace-element variability have with sea-salt and biomass burning apportionment. The lidar backscatter profiles can provide information about the vertical distribution of aerosol layers. By combining those data with the air mass trajectories, it is possible to infer the local sources and transport times for the aerosol plumes to arrive at the SPMA. The retrieval of the

aerosol optical properties is based on the measurements of the light backscattered by aerosols, which allows us to obtain the aerosol backscatter coefficient ($\boldsymbol{\beta_{aer}}(\boldsymbol{\lambda},\boldsymbol{z})$, where $\beta_{aer}$ is the aerosol backscatter and $z$ is the altitude) at 532 nm, from 300 m up to an altitude of 5–6 km. The determination of the vertical profile of the aerosol backscatter and extinction coefficients ($\boldsymbol{\alpha_{aer}}(\boldsymbol{\lambda},\boldsymbol{z})$) relies on the lidar inversion technique following a modified Klett algorithm (Klett, 1981; 1983) under the assumption of the single scattering approximation. To solve the lidar equation requires establishing a relationship between

$\boldsymbol{\alpha_{aer}}(\boldsymbol{\lambda},\boldsymbol{z})$ and ($\boldsymbol{\beta_{aer}}(\boldsymbol{\lambda},\boldsymbol{z})$. This is typically achieved by assuming that the aerosol extinction-to-backscatter ratio is independent of altitude that $\boldsymbol{R}(\boldsymbol{\lambda}) = \boldsymbol{\alpha_{aer}}(\boldsymbol{\lambda},\boldsymbol{z})/\boldsymbol{\beta_{aer}}(\boldsymbol{\lambda},\boldsymbol{z})$. However, it is known that the lidar ratio (LR) depends on various physical-chemical parameters inherent to the aerosols being inspected, such as the aerosol refractive index and the size/shape distributions of the aerosol particles (Liou, 2002). To derive the appropriate values of the vertical profile of aerosol backscatter coefficient in the lower troposphere, we use an iterative inversion approach that tunes the LR values

based on the inter-comparison of the aerosol optical depth (AOD) values derived from the lidar and collocated Aerosol Robotic Network sunphotometer data (Marenco et al., 2002) or in some cases from Aqua and Terra Satellite Moderate Resolution Imaging Spectroradiometer (MODIS) data (Remer et al., 2005). Once the values of the vertical profile of the aerosol backscatter coefficient have been derived (when the difference between the sunphotometer-derived and lidar-derived AODs was less than 10%), we reapply the Klett inversion technique using the appropriate LR values to retrieve the final

values of the vertical profiles of the backscatter and extinction coefficient at 532 nm. The vertical profiles of pressure and temperature measured by radiosondes launched twice a day, at 1200 and 0000h (local time), at a distance of about 10 km from where the MSP-Lidar I system is located, were used to obtain the molecular contribution based in the using the Bucholtz approach (Bucholtz, 1995). The molecular contribution is calculated in an altitude range with a negligible particle load, being considered an aerosol-free region, the altitude range set here being 8–12 km.

Air masses trajectories are computed using the Global Data Assimilation System (GDAS), an operational system from the National Weather Service National Centers for Environmental Prediction. The GDAS is used as an input in the HYSPLIT model (version 4.8). Forward or backward trajectories are selected on a case-by-case basis, and the model vertical velocity option is chosen (Draxler and Hess, 1998). These trajectories provide the source of air masses arriving at the MASP at different levels (altitudes) and preset time intervals, the coordinates of the IAG, where the instruments are installed, constituting a starting point.

## 3 Results and discussion

### 3.1 Meteorological parameters, CCN and PNC

This study was conducted in the winter, when the climate is mostly dry, although three rain events were observed during the study period. Meteorological parameters, PNC and CCN counts are presented in Table 1. As can be seen, the mean temperature was highest on day 5, whereas it was lowest on days 7 and 8, the greatest variation in temperature (16.1°C) occurring on day 11. The mean relative humidity (RH) was above 70% on all days except days 4 and 5, when it was 62% and 58%, respectively. The greatest variation in RH occurred on days 3, 4 and 11. Precipitation events were observed in the morning, on day 6, and at night, on days 11 and 13 (Fig. 2a). Figure 2 shows hourly variations in meteorological variables (temperature, RH and precipitation), CCN and PNC. The PNCs were highest between 0700 h and 1900 h, whereas they were lowest between 1900 h and 0700 h (Fig. 2b), hereafter referred to as the diurnal and nocturnal periods, respectively. During the diurnal period, PNCs were elevated, especially during the rush hours, primarily associated with vehicle emissions. However, CCN peaks showed the opposite behaviour, CCN concentrations being higher during the afternoon and the nocturnal period. These patterns are discussed further in section 3.4. As expected, periods of precipitation were characterised by lower PNCs and CCN concentrations. To eliminate possible interference with the data analysis, we excluded CCN and PNC values for the periods of precipitation. The diameter peaked at less than 100 nm and presented a mean value of approximately 55 nm for all days. The mean peak diameter was highest (92 nm) on day 6 and lowest (29 nm) on day 14, characterising Aitken mode (25-100 nm) (Dal Maso et al., 2005). The mean PND was 44 nm and 62 nm for the diurnal and for nocturnal periods, respectively, the difference being attributed to the high PNCs during the morning hours. The particle size distributions and NPF events are discussed in section 3.2.

To compare our PNC and CCN values with those of other studies, we plotted our results against the results of recent studies conducted in other urban regions (Fig. 3, Table S1 on Supplementary Information). The PNCs were higher during the diurnal period than during the nocturnal period, whereas, CCN concentrations were comparable between the two periods. The higher PNCs during the diurnal period were expected, given the increased emission of pollutants from local sources such as vehicular traffic. However, the fact that CCN concentrations did not vary significantly between the nocturnal and diurnal periods indicates that CCN formation was more efficient during the nocturnal period. In a study conducted in Beijing, Gunthe et al. (2011) reported similar behaviour for fresh pollutant emissions and regional aged pollution, the latter

presenting higher efficiency for CCN formation, as evidenced by lower PNCs and higher CCN concentrations. Those observations are supported by Köhler theory predictions, related to the greater efficiency of larger particles in CCN formation, which is extensively discussed in section 3.4.

The overall mean PNC and CCN values obtained in the present study were similar to those observed by Almeida et al. (2014) for the MASP during October 2012. However, our PNC values were lower than those reported for the MASP by Backman et al. (2012) for October 2010 and January 2011. This variability can be attributed to different meteorological conditions, seasonal differences and the decrease in $SO_2$ emissions associated with the recently mandated reduction in sulphur concentrations in diesel fuel (Kumar et al., 2016; CETESB, 2015). In a study conducted in Shanghai, Leng et al. (2013) reported PNC values similar to those obtained for the SPMA in the present study, although the CCN concentrations reported by those authors were higher; that might be related to the coastal environment, which increases the concentrations of most soluble compounds, such as ionic species ($SO_4^{-2}$, $NO_3^-$, $Na^+$, $Cl^-$, $K^+$), in the aerosol chemical composition. Our results showed PNC values similar to those observed for London and Madrid (Reche et al., 2011; Gómez-Moreno et al., 2011, respectively). In these three urban areas (London, Madrid and the MASP), transport emissions constitute the main pollution source and there are light industries around urban regions. London and Madrid have higher population densities than does the MASP. However, the vehicle fleet in the SPMA is larger than is that in any of the other urban regions evaluated, although Madrid has the highest vehicle/inhabitant ratio and the highest proportion of diesel-powered vehicles (~50%). With a population of over 20 million, Mexico City is the largest megacity in North America. Although comparable to the MASP, the mean PNC for Mexico City in 2006 was double that reported for the 2012–2014 period in the MASP (Kalafut-Pettibone et al., 2011; Almeida et al., 2014; This study). Nevertheless, the mean PNC reported for Mexico City was similar to that observed for 2010 in the MASP (Backman et al., 2010). As previously mentioned, the lower PNCs in the MASP can be attributed to legislation that mandated a reduction in the concentration of sulphur in diesel fuel. The CETESB reported a ~10% reduction in the emission of particulate matter from diesel-powered vehicles between 2010 and 2015. In the MASP, such vehicles emitted 26% of all particulate matter attributed to anthropogenic sources during 2015 (CETESB 2016). In the case of Mexico City, 50% of all particulate emissions in 2006 were from diesel-powered vehicles (Kalafut-Pettibone et al., 2011).

## 3.2 Diurnal PND variability and NPF events

Figure 4a shows the variability in UV radiation intensity, as well as in the concentrations of $SO_2$, $O_3$, NO and $NO_x$. The gaseous concentrations measured at the four CETESB stations were selected according to the direction of the prevailing winds. For the most part, the diurnal periods presented high UV intensity (~30 W m$^{-2}$) and $O_3$ concentrations (80–100 µg m$^{-3}$). On days 4, 5, 6, 12 and 14, NO and $NO_x$ concentrations were elevated (100–420 ppb), which was mainly associated with vehicle emissions and prevailing winds from the north-west and north-east. The Marginal Pinheiros and Marginal Tietê roadways are approximately 1 km and 4 km, respectively, from the sampling site. In 2014, approximately 80% of total $NO_x$

emissions in the MASP were attributed to vehicular traffic (CETESB, 2015). Figure 4a also shows that the higher $O_3$ concentrations were preceded by $NO_x$ peaks. This pattern was expected, given the chemical reactions that occur between $NO_x$ and VOCs in the presence of UV radiation, which induces photochemical activity and the formation of secondary pollutants such as $O_3$ and SOAs (Monks et al., 2015).

Sulphate and nitrate compounds, which are formed by $SO_2$ and $NO_x$ precursors, are important components of the nucleation process. The $SO_2$ concentrations measured at CETESB stations were plotted according to wind direction in order to represent air masses arriving at the measurement site (Fig. 4a). The highest concentrations of $SO_2$ were associated with winds from the south-east (Congonhas station), whereas winds from north were associated with lower $SO_2$ concentrations. To the south-east, there are two large industrial areas that represent an important source of $SO_2$ (Fig. 1). During 2014, industrial activities

accounted for 56.9% of the sulphur oxides emitted in the MASP. The $SO_2$ transport from industrial areas is supported by some studies, which showed that marine air masses and industrial emissions from Cubatão reached the MASP. Sea-salt transport is discussed further in section 3.3, based on the chemical and trajectory analyses.

In order to identify NPF events, we adopted the following requirements described by Dal Maso et al.(2005), which have been extensively employed in other studies (Backman et al., 2012; Betha et al., 2013; Ma and Birmili, 2015): a distinctly

new particle mode (nucleation mode, particle diameters <25 nm) must appear in the size distribution; and the new mode must present a duration of several hours and show growth over time. These requirements were complemented by Ma and Birmili (2015), who stated the following: a sudden increase in PNC, above nocturnal background concentration, must occur in nucleation mode; the burst must be sustained for a minimum of 1 h; and a decrease in the concentration of the nucleation mode particles is expected at end of day.

Dal Maso et al.(2005) identified two classes of NPF events, the first of which is divided into two subclasses: (Ia) a clear and pronounced particle formation event, followed by particle growth; (Ib) a particle formation event in which particle growth can be observed from the nucleation, although without a clear nucleation burst; and (II) a particle formation event in which particle growth from nucleation mode cannot be determined.

The contour of the PNDs was plotted in order to identify and classify NPF event days (Fig. 4c). An NPF event meeting all of

the requirements described above was observed on day 12. The nucleation burst presented a duration of 3h, followed by particle growth until the beginning of the nocturnal period, when a brief decrease in the concentration was observed. An NPF event occurring on day 14 was classified as class II because there was no significant particle growth after nucleation. That event presented nucleation bursts similar to those observed in Singapore by Betha et al. (2013) and in Brisbane, Australia, by Cheung et al. (2011). Those authors associated the events with winds from industrial areas, which carried large quantities of

$SO_2$ and VOCs. There were also NPF events on days 7, 8 and 10, all of those events being categorised as class Ib, clearly showing particle growth throughout the diurnal period. On those days, moderate to high concentrations of $SO_2$ (5–20 µg m$^{-3}$) were observed at the Congonhas station, which suggests that the nucleation process began near that station, followed by the detection of particle growth at the USP station. Backman et al. (2012) observed similar class Ib events in the MASP.

The nucleation of primary aerosol occurs near local sources such as vehicle and industrial emissions. No nucleation events of

that type were detected in the present study because of the distance between the sources and measurement site. However, five different secondary aerosol NPF events were identified. The class Ia and II events were mostly associated with high concentrations of $NO_x$ and $O_3$, as well as with photochemical activity, whereas the class Ib events were characterised as NPF passing through the measurement site only after the initiation of the event. The impact that those NPF events have on the AR (Fig. 4d) and on $D_{act}$ are discussed in section 3.4.

As previously mentioned, $O_3$ plays a fundamental role in SOA formation via VOC oxidation, its concentrations being indicative of the efficiency of the photochemical process (Sorribas et al., 2015). However, after the nucleation process, SOAs drive particle growth to larger sizes, primarily by condensation of non-volatile molecules (Pierce et al., 2012; Donahue et al., 2013). In addition, the particle growth rate is the most important factor in determining the extent to which new particles become CCN during NPF events (Leng et al., 2013). As can be seen in Figs. 4a and c, the NPF events observed on days 7 and 8 occurred at low $O_3$ concentrations, whereas those observed on days 10, 12 and 14 occurred at high $O_3$ concentrations. To assess the importance of photochemical activity and SOA production to particle diameter and to the AR, we plotted NPF events under low and high $O_3$ concentrations (Fig. S1, on Supplementary Information). As expected, particles formed during low-$O_3$NPF were smaller than were those formed during high-$O_3$NPF (Fig. 4c). In addition, the AR was higher for the particles formed during high-$O_3$NPF than for those formed during low-$O_3$NPF. That is in agreement with the findings of studies predicting or demonstrating the efficiency of SOA condensation in inducing particles to become CCN (Pierce et al., 2012; Riipinen et al., 2011)

We drew PND contour plots in order to evaluate particle nucleation and growth progression during NPF events of the various classes and even on non-NPF event days. The proportional distribution of hourly PNCs in three modes (nucleation, Aitken and accumulation (100-1000 nm)) was also plotted, as were the hourly PNDs (Fig. 5). The NPF event on day 12 showed a nucleation burst between 1000 h and 1500 h (Fig. 5a), which was confirmed by the fact that the proportion of particles in nucleation mode was higher (>50%) during that period than at any other time on that day. The particle growth in subsequent hours was accompanied by an increase in the proportion of particles in Aitken mode, as can be noted in the hourly PNCs and PND (Fig. 5a).

The class Ib NPF events showed characteristic particle growth after midday, with probable occurrence of a nucleation event, which was not observed between 1000 h and 1200 h. That particle growth was also detected by a higher PNC (63%) of Aitken mode at 1500 h and an increase in peak diameters between 1200 h and 1500 h (Fig 4b). It is noteworthy that the proportion of particles in Aitken was high (>45%) for most hours on the three selected days, which indicates a continuous increase in the size of primary and new particles. This observation is supported by the fact that the local traffic presents constant emissions, with high density during the diurnal period and low density during the nocturnal period, which agrees with the total PNC values shown in the hourly frequency graphics (Fig. 5).

## 3.3 Apportionment of biomass burning and sea-salt from remote sources

A combination of lidar, HYSPLIT trajectory and size-distributed chemical composition (Na, Ca, Ti, K, Cl P, Fe, Mn, Pb, Cu,

Zn, S and BC) analyses were used in order to apportion the contribution of sea-salt and biomass burning within the air masses arriving at the MASP. Over the past 30 years, various studies conducted in the MASP have employed receptor modelling and aerosol chemical composition analysis in order to identify the main sources of atmospheric pollutants (Bouéres and Orsini, 1981; Andrade et al., 1994; Castanho and Artaxo, 2001; Sanchez-Ccoyllo and Andrade, 2002; Sanchez-Ccoyllo et al., 2008; Andrade et al., 2012). Those studies have determined that vehicle emissions account for 50–60% of fine particles, thus constituting the main source, followed by oil boilers (accounting for 20–40%), road dust (accounting for 10–30%), industrial emissions (accounting for 10–20%), biomass burning (accounting for 10–20%) and construction activities (accounting for ~10%). Some studies carried on in the MASP evaluated the elemental profiles of the principal urban pollution sources, characterising observed elements such as Mn, Pt, Ni, Cu, Pb, Cr and Zn as markers of gasoline and alcohol emissions (Silva et al.,2010). Diesel burning by heavy-duty vehicles is associated mainly with BC and S. Suspended road dust, characterised inside and outside road tunnels, has been found to be comprise BC, Si, Al, Fe, Ca, Mg, K, Ti and S, which denotes a mixture of soil, pavement abrasion, tire wear, brake wear and vehicular emission (Hetem and Andrade 2016). In a general approach, other studies found BC to be related to biomass burning; Na and Cl to be related to sea-salt contribution; and Fe, Cu, Zn, Cr, Pb and Ni to be related to industrial emissions (Bzdek et al., 2012; Calvo et al., 2013; Taiwo et al., 2014).

Local pollution sources presents acontinuous contribution to atmospheric aerosol, functioning as background to MASP aerosol, emission of which are higher during the diurnal period and lower during nocturnal period, as demonstrated by the PNC variability (Fig. 2). The objective of this study was not to discriminate among the main local sources but rather to identify periods during which remote sources were detectable, in order to evaluate how CCN activation properties are affected by such sources, in association with local sources, which cannot be excluded during urban aerosol measurements.

Figure 6 shows the mass size distributions of trace-elements from MOUDI membrane samples for different periods. The concentrations of BC were high (~0.9%) on days 6, 12 and 13 (Fig. 6a), whereas they were intermediate (~0.4%) on days 7, 8, 9 and 14. The concentrations of Cl, K and Pb were high on days 8 and 9 (Fig. 6b–d), and the concentration of K was intermediate (~0.2 $\mu g \, cm^{-3}$) on day 6. The elements Ti, Fe, Ca and Cu showed intermediate concentrations for most days, with peaks in the morning on day 6 and in the evening on day 3 (Fig. 6e–h).

Figures 7c and 7d present the vertical profile for two measurement days obtained with the MSP-Lidar I system. On day 6, several thin aerosol plumes were detected above the planetary boundary layer (PBL). According to the range-corrected lidar signal (Fig.7c), most of the aerosol load in the atmosphere is concentrated inside the PBL, approximately 500 m above ground level (AGL). However, several aerosol layers can clearly be seen above the PBL, at 1000–2000 m AGL, and a thin aerosol layer can be seen at approximately 3000 m AGL. The mean aerosol backscatter profile at 532 nm (Fig.7c), retrieved applying the Klett inversion method, as previously described, showed a peak of $0.006 \, km^{-1}sr^{-1}$ at approximately 500 m AGL, representing all aerosol trapped inside the PBL, and another peak of $0.0018 \, km^{-1}sr^{-1}$ at approximately 2800 m AGL. The mean aerosol vertical profile at 532 nm clearly shows a backscatter signal from aerosol plumes up to 4000 m AGL, which would typically be from a source outside the SPMA.

When the lidar data were analysed by tuning the LR values based on the inter-comparison of the MODIS-derived AOD, a mean lidar ratio of 70±14 sr was obtained, which is within the range associated with absorbing particles from biomass burning (Müller et al., 2007; Baars et al., 2015). The air mass backward trajectories simulated using the HYSPLIT model (Fig. 7a) strengthens the hypothesis that aerosol from biomass burning was transported from the northern and central-west portion of Brazil, where there were several fire focus, reaching higher levels and being transported to the MASP at altitudes of 3000–4000 m AGL. The same pattern was found for day 12 (Fig. 7d), when thin aerosol plumes were also detected above the PBL, at 2400–4000 m AGL, as can be seen in the mean backscatter profile (Fig. 7d). The mean lidar ratio for this case was also 70±14 sr, again associating the plume with a biomass burning aerosol type. Air masses trajectories (Fig.7b) indicated that plumes detected at the MASP by the lidar system had been transported from the central-west portion of the country and had travelled across regions with several fire focus. The lidar analysis and measurements were in agreement with the previous analysis of size distribution of elemental concentrations in samples collected by MOUDI on days 6 and 12, when high concentrations of BC were observed.

Previous studies have shown the contribution of sea-salt to the aerosol profile in the MASP (Andrade et al., 1994; Castanho and Artaxo, 2001; Vara-Vela et al., 2016). As can be seen in Fig. 4b, the prevailing wind direction was from the South-East during the nocturnal period on days 7 and 8, which is in line with the low-altitude air mass trajectories shown in Fig. S2 in the Supplementary Information; the higher concentrations of Cl indicate sea-salt transport from the coastal region, as evidenced by the increased Pb concentrations on those days. Normally, Pb is linked to emissions from anthropogenic sources, mainly industrial emissions, as the contribution from fuel burning has decreased substantially after the ban on Pb in gasoline fuels. Using Pb isotopic signatures, Gioia et al. (2010) demonstrated by aerosols from the Cubatão industrial area arrive at the MASP when south-east wind predominates. As can be seen in Fig. 1, Cubatão is located in the coastal region, corroborating the sea-salt transport. Figure 8c shows the range-corrected signal of MASP MSP-Lidar I system in the atmosphere throughout day 4. Between 2030h UTC and 2213h UTC, signal attenuation was detected at 500–1000 m AGL, which is associated with the intrusion of clean air at the PBL of the MASP. Typically, such intrusion is related to sea breeze transported to MASP (Rodrigues et al., 2013). The mean backscatter profile showed the presence of aerosol only within the PBL, where a smaller aerosol backscatter signal could be seen between 300 m and 1200 m AGL, associated with the clean air region inside the PBL. The mean lidar ratio for this measurement was 40±8 sr, which is within the range associated with the urban aerosol type (Baars et al., 2015).

In summary, sea-salt air masses arriving at the MASP were observed during the nocturnal period on three of the days evaluated. During the nocturnal period of days 4 and 7, 8, sea-salt events were observed by Lidar and trace-element concentration analysis, respectively. Throughout the year, sea breezes arrive at the MASP in the afternoon and evening (Oliveira et al., 2002; Freitas et al., 2007). In the present study, plumes generated from biomass burning were detected by lidar on days 6 and 12, being associated with an increase in BC on those specific days. In Brazil, numerous biomass burning events occur every year from July to November, mainly in the central and northern regions of the country. However, many such events, associated with agricultural activities, occur within the state of São Paulo throughout the year (Kumar et al.,

2016). All focus fire, as shown in figures 7a and 7b, were identified from Geostationary Operational Environmental Satellite images (GOES).

### 3.4 Effects of local sources, remote sources and NPF events on CCN activation properties

The AR is calculated as the ratio between CCN and PNC, which is described in the literature as "CCN efficiency" or "CCN fraction". This shows the fraction of total particles that are able to produce CCN. The AR value has been extensively discussed in several papers, which investigated the variability of CCN properties under different conditions and during diverse pollution events (Table 2). These bulk ratios do not consider aerosol particle size, which is very important to CCN activation. The Köhler theory predicts that critical supersaturation will decrease with increasing size of a soluble particle, or increase mass of soluble substance in a particle containing both soluble and insoluble substance (McFiggans et al., 2006; Andreae and Rosenfeld, 2008).

The mean AR values (at the various levels of supersaturation) were 0.22±0.05 (at 0.2%), 0.33±0.09 (at 0.4%), 0.41±0.05 (at 0.6%), 0.48±0.09 (at 0.8%) and 0.53±0.09 (at 1.0%), the nocturnal means being twice as high as the diurnal means for all supersaturation levels evaluated (Table 2 and Fig. 9a). In addition, the PNC was five times higher during the diurnal period than during the nocturnal period. In order to determine the effects that NPF events have on ARs, the mean values were split into diurnal and nocturnal periods with and without NPF or remote source events (non-event days), as seen in Fig. 9a. We observed a tendency for ARs to be lower during diurnal periods with NPF than during those without, although we observed no such tendency for nocturnal periods. That is in agreement with the findings of previous studies suggesting that larger particles act more efficiently as CCN than do smaller particles (Sihto et al., 2011; Dusek et al., 2010). In a study conducted in Finland, some events featured NPF bursts, followed by particle growth. During those events, the ARs were very low, due to higher PNCs and smaller particle sizes (Sihto et al., 2011). In the present study, the AR increased substantially after each NPF event, although it was still lower than that observed on the non-event days. In addition, we observed increase in the number of CCN during the nocturnal periods after NPF events, as showed in Supplementary Information (Fig. S3), attributed to the high rate of growth among the particles formed during NPF, an observation that is supported by previous studies (Sihto et al., 2011; Yue et al., 2011; Peng et al., 2014).

Figures 4c and 9b show the predominance of larger particles during the nocturnal period, as a result of primary and secondary aerosol growth during the diurnal period. The primary aerosols are emitted mainly during the morning rush, whereas secondary aerosols are formed, by nucleation, near midday, as previously discussed. Therefore, the higher AR during the nocturnal period was an expected finding, given the efficiency of larger particles to act as CCN, together with the lower PNC. During the diurnal period, the mean AR values were similar to those observed in other urban areas, although not to those observed in coastal areas (Leng et al., 2013; Furutani et al., 2008), as indicated by recent studies of fresh urban pollution conducted in the MASP, Vienna and Beijing (Almeida et al., 2014; Burkartet al., 2012; Gunthe et al., 2011). However, the AR reported for Beijing was twice that found for the MASP in the present study, considering entire campaign

for both, although the PNC values were similar. In addition, the AR values observed for the SPMA in the present study are comparable to the fresh ship exhaust emissions reported in a study conducted along the coast of California (Furutani et al., 2008). The nocturnal AR values observed for the MASP were similar to those reported in a study conducted in a forest environment (Sihto et al., 2011) and in the coastal environments, although opposite those observed in others urban

environments (Table 2). However, the mean nocturnal AR and PNC values were higher for aged pollution in Beijing than for the MASP.

Figure 9b shows the hourly mean AR values for days with and without NPF. The AR values were lowest between 1000 h and 1500 h, whereas they were highest during the nocturnal period. The AR began to decrease after 0600 h and subsequently began to increase after 1500 h. In the MASP, the morning rush occurs between 0700 h and 1000h, whereas the afternoon

rush occurs between 1700 h to 2000 h. The decrease begin and the lower values of AR occurred during the morning rush and formation of a secondary aerosol. Vehicle emissions and secondary aerosol formation constitute the main sources of particles in megacities such as the MASP (Andrade et al., 2012). As previously discussed, the formation of secondary aerosols is favoured by photochemical reactions, which depend on solar radiation (UV and visible wavelength light), $O_3$ and hydroxyl radicals. Figure 4a shows UV radiation and $O_3$ peak at midday, after which there is secondary aerosol formation and growth

as demonstrated by intermediate and high concentrations in nucleation and Aitken modes, respectively, in the afternoon (Fig. 4c). Therefore, vehicle emissions and secondary aerosol formation probably present the major contribution to lower AR values during the diurnal period.

The variability in $NO_x$ and CO concentrations measured at the Marginal Tietê is mainly associated with vehicular traffic, because fuel combustion is the predominant source of such pollutants in the MASP. As can be seen in Fig. 9b and Fig. 10c,

an increase in $NO_x$ occurred during the morning rush and was clearly related to the decrease in AR and particle diameter mode (Fig. 9b), as well as to the increase in $D_{act}$ (Fig. 10c). The $NO_x$ decrease after midday was not associated with the decrease in traffic volume but with photochemical reactions and secondary formation of pollutants, as previously discussed. We believe that the $NO_x$ and $O_3$ variability corroborates our assumption, explaining the decrease in AR values during the diurnal period in the presence of high vehicular traffic volume and secondary aerosol formation.

The increase in AR after 1500 h can be correlated with the increase in particle diameter and CCN concentrations. In addition, the production of SOAs by VOC degradation can result in the formation of polar functional groups that contribute to the growth and hydrophilicity of particles. Therefore, aged aerosols emitted and formed during the morning from vehicle emissions and secondary particles mainly showed an increase in AR at the end of the afternoon and during the nocturnal period. This observation is in agreement with those of previous studies in which increased CCN activation was associated

with aged anthropogenic emissions (Quinn et al., 2008; Furutani et al., 2008; Dusek et al., 2010; Bougiatioti et al., 2011; Zhang et al., 2014). Nevertheless, vehicle emissions from afternoon rush (1700–2000 h) had less effect on the AR than did those from the morning rush, probably due to the absence of secondary aerosols and mixing with aged aerosols. The hourly particle diameter mode clearly shows the effect of particle size on AR values, lower ARs being accompanied by smaller particle size modes (Fig. 9b). Our observation is in agreement with those of previous studies demonstrating the strong effect

of particle size on CCN activity.

Variability in $D_{act}$ reflects the effect of aerosol chemical composition, associated with hygroscopicity or water activity. Higher number of the hygroscopic particles are activated at small diameters, whereas hydrophobic particles are typically activated at large diameters (Sihto et al., 2011). Frank et al. (2006) observed that, at a supersaturation of 0.4%, $D_{act}$ values were 50 nm for pure sulphate and marine aerosol; 70 nm for continental aerosol, represented by regional pollution; and 125 nm for fresh biomass smoke aerosol. If the chemical composition of the particles is homogeneous, the apparent activation diameter ($D_{act}$) can be calculated by integrating size-distributed aerosol from larger particles toward smaller ones until the integral equals CCN concentration at a given supersaturation (Bulkart et al., 2011). However, atmospheric aerosol presents a variable and complex composition in terms of size distribution, containing different proportions of soluble and insoluble compounds. Therefore, this must be taken into account in order to discuss the topic adequately.

The overall mean $D_{act}$ varied between 54 nm (at SS = 1.0%) and 155 nm (at SS= 0.2%) for the nocturnal and diurnal periods, respectively. A comparison between the $D_{act}$ values obtained in this study and those reported in the literature is shown in Table S2. As can be seen in Fig. 10a, $D_{act}$ presents non-linear decrease as function of increase in supersaturation. The $D_{act}$ was higher during the diurnal period than during the nocturnal period, as was expected given the AR data. The mean $D_{act}$ values for diurnal period with biomass burning and NPF events were similar to those observed for non-event days. The $D_{act}$ values for nocturnal periods after NPF or during sea-salt events were similar to those observed after non-event days, although the $D_{act}$ values were slightly higher for nocturnal periods during which there were biomass burning plumes, mainly when the SS < 0.6%. At high supersaturation values, particles with different chemical composition and therefore hygroscopicity have only a weak effect on CCN activity (Sihto et al., 2011; Zhang et al., 2014).

Figure 10c shows the time series of the mean $D_{act}$ values for days with and without NPF events. The $D_{act}$ values recorded during rush hours were higher on days with NPF than on those without. A number of hours before the occurrence of NPF events in which lower PNCs and larger particles predominated (Fig. 4c and Fig. 10c), lower AR values and higher $D_{act}$ values were observed (Fig. 9b and Fig. 10c, respectively). This observation could be related to a possible increase in SOA precursor compounds in condensed phases before NPF events. Some of those compounds are monoterpenes and aromatic compounds, which have been reported to present low hygroscopicity (Kroll and Seinfeld, 2008). Except during the above mentioned pre-NPF periods, days with and without NPF events showed similar nocturnal means.

The efficiency of aerosol particles to act as CCN can be estimated on the basis of AR and $D_{act}$ data. The AR is dependent on particle size and chemical composition, whereas $D_{act}$ is dependent on chemical composition only (Furutani et al., 2008). As can be seen in Fig. 10b, the non-linear correlation between AR and $D_{act}$ can be related to different chemical composition and size distribution of aerosol. During the diurnal period, the $D_{act}$ was increased and the AR was decreased, whereas the inverse was true for the nocturnal period. In general, the diurnal period is associated with particles that are less hygroscopic and smaller, mainly emitted by vehicular traffic. However, the decreased $D_{act}$ and increased AR were observed in the nocturnal period, being associated with larger and more hygroscopic particles. Our observations support the assumption that nocturnal samples typically comprise greater concentrations of water soluble species, such as $(NH_4)_2SO_4$, SOA, $NO_3^-$, and of marine

air than do diurnal samples. Our findings are also in keeping with those of other studies showing that aged aerosols present high hygroscopicity (Gunthe et al., 2011; Bougiatioti et al., 2011).

We compared the $D_{act}$ values obtained in this study with those reported in previous studies (Fig. 11 and Table S2). Our nocturnal means were similar to those reported for continental air masses over Shanghai (Leng et al., 2013), for the

atmosphere over a boreal forest in Finland (Sihto et al., 2011) and for periods without air masses containing anthropogenic emissions over California (Furutani et al., 2008), as detailed in Table S2, on Supplementary Information. Our diurnal means were comparable to those reported for urban/industrial and fresh polluted air masses in Mexico and California (Quinn et al., 2008), even the differences being less than statistically significant (Fig.11). The mean $D_{act}$ values for non-event nocturnal periods and during sea-salt events were comparable to those observed for the Atlantic and Pacific Oceans, as well as for the

Gulf of Mexico and for aged anthropogenic emissions in California (Furutani et al., 2008; Quinn et al., 2008)

On the basis of our data for diurnal and nocturnal periods with and without NPF and remote source events, we calculated the mean AR and $D_{act}$ for specific periods (Fig. 12c and Fig. 12d), in order to identify the influence that different events have on CCN activation. The CCN activation properties related to event and non-event days were calculated for the diurnal and nocturnal periods. The variability for diurnal and nocturnal periods was taken into account, and the events were calculated

for each period. In addition, we calculated the mean AR and $D_{act}$ values for nocturnal periods after NPF and non-NPF days, in order to evaluate the effects of NPF on CCN activation. Another important information is the variability in the volume of vehicular traffic, as estimated from the mean $NO_x$ and CO values (Fig. 12b), which was previously associated with decreased CCN activation. The overestimation of the volume of vehicular traffic can result in the underestimation of effects of remote source events on CCN activation.

As can be seen in Fig.12c, diurnal periods with NPF showed lower AR values than did biomass burning and non-event days. However, AR values were similar among nights after NPF with those non-event and sea salt influence nights and also biomass burning after NPF events. The nocturnal increase in AR after NPF events was related to particle growth (Fig. 12c). Lower AR for nocturnal periods with sea-salt influence after NPF can be attributed to smaller particle size, whereas sea-salt events in these nocturnal periods occurred after low-$O_3$NPF events (on days 7 and 8). As previously discussed, low-$O_3$NPF

events were characterised by the formation of particles that were smaller than those formed during high-$O_3$NPF events. The effect of another sea-salt event (day 4) on CCN activation might have been underestimated due to high traffic volume during the nocturnal period, as indicated by high concentrations of $NO_x$ and CO. The AR values were slightly lower during diurnal and nocturnal periods with biomass burning plumes only (i.e. without NPF events) than during those with non-events. In addition, during the nocturnal periods, the biomass-burning events promoted slightly higher $D_{act}$ values than did NPF and sea

salt, which is attributable to lower particle hygroscopicity. Although there was a lack of statistical significance, we observed a tendency for biomass burning affected air masses arriving at the MASP to decrease the activation properties.

# 4 Summary and conclusions

The activation of CCN is quite important to cloud formation and consequently to the hydrological cycle. The aim of this study was to evaluate the variability in CCN activation parameters under the influence of local sources, remote sources, and NPF, in order to assess the impact of those events on CCN activity in the MASP. The regional atmosphere is highly affected by local sources, with a major contribution from vehicle emissions, as well as by remote sources such as biomass burning and sea-salt transport.

During the diurnal periods, PNCs were higher and particle sizes were smaller than during the nocturnal periods, a finding that was attributed to fresh vehicle emissions and NPF, whereas the lower PNCs and larger particle sizes observed during the nocturnal periods were attributed to the ageing and post-growth deposition of particles (Fig. 12a). Five NPF events were identified, four being of the class I type. In addition, NPF events were found to have a significant effect on particle size distribution. Biomass-burning and sea-salt air masses were identified by a combination of lidar analysis, HYSPLIT trajectory analysis and size-distributed chemical composition.

The AR and $D_{act}$ CCN activation parameters presented a clear pattern for the diurnal and nocturnal periods, the diurnal period featuring higher $D_{act}$ and lower AR. The size distribution made a marked contribution to that pattern, according to the assumption that larger particles are more efficient for CCN activation than are smaller particles. That hypothesis is supported by AR and $D_{act}$ differences between days with and without NPF events. Slight differences for activation parameters could be observed among sea-salt and biomass-burning influenced air masses, as well as non-event nocturnal periods (Fig. 12c and Fig.12d). Those slight differences can be explained by complex mixtures between aerosols from local and external sources, the latter travelling longer distances before reaching to the measurement site. Dilution, mixing and ageing of aerosols from remote sources occur during travel from the source emission area to the measurement site. The $D_{act}$ calculated for sea-salt air masses in this study were comparable to that observed over the Atlantic and Pacific Oceans. In summary, local pollution sources, represented mainly by vehicular traffic emissions, had greater negative effects on activation parameters than did remote sources.

This was the first study to evaluate the effects of remote sources and NPF events on CCN activation properties in urban area on South America and only the second to assess CCN variability in the MASP. Our results show the influence that local emissions, long-range transport of sea-salt, biomass-burning plumes and NPF events have on CCN properties. Therefore, long-term studies covering different seasons and sampling site environments (e.g. coastal, urban and rural) are needed in order to represent CCN variability more accurately under diverse environmental conditions. To improve understanding of the influence that aerosol composition and provenance ehave on CCN activity, real-time monitoring of aerosol chemical composition and CCN size distribution is also necessary.

## Acknowledgements

This work received financial support as part of the University Global Partnership Network (UGPN) for the "Emissions and

role of fine aerosol particles in the formation of clouds and precipitation (eRAIN)" project grant, awarded jointly to Universities of Surrey and São Paulo. This study is also part of a major project entitled "Narrowing the Uncertainties in Aerosol and Climate changes in the state of São Paulo (NUANCE)", funded by the *Fundação de Amparo à Pesquisa do Estado de São Paulo* (FAPESP, São Paulo Research Foundation). Carlos Eduardo Souto-Oliveira is the recipient of a PhD scholarship from the Brazilian *Coordenação de Aperfeiçoamento de Pessoal de Nível Superior* (CAPES, Office for the Advancement of Higher Education).

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

**Table 1.** Diurnal data for meteorological parameters (temperature, relative humidity and rain), particle number concentration and cloud condensation nuclei during the period evaluated (19 August to 3 September, 2014). Peak diameter was calculated by size distribution mode. The last column, represent periods with NPF events and remote sources influence on MASP atmosphere.

| Day | | Date | Temperature (°C) | RH (%) | Rain | PNC (cm⁻³) | | Dp (nm) | Cloud Condensation Nuclei (mean ± SD) | | | | | Remote Source* and NPF | |
|---|---|---|---|---|---|---|---|---|---|---|---|---|---|---|---|
| | | | mean (range) | mean (range) | (mm) | mean ± SD | (range) | | SS = 0.2% | SS = 0.4% | SS = 0.6% | SS = 0.8% | SS = 1.0% | Diurnal | Nocturnal |
| 1 | Tue | 8/19/14 | 15 (11 - 20) | 83 (64 - 96) | 0.0 | 7654 ± 4582 | 1874 - 16607 | 50 | 1688 ± 569 | 1809 ± 624 | 2270 ± 830 | 2555 ± 965 | 2811 ± 1029 | | |
| 2 | Wed | 8/20/14 | 16 (11 - 23) | 80 (43 - 98) | 0.0 | 13086 ± 8242 | 3335 - 35398 | 45 | 2240 ± 780 | 2385 ± 961 | 2969 ± 1191 | 3431 ± 1371 | 3770 ± 1611 | | |
| 3 | Thu | 8/21/14 | 18 (13 - 27) | 70 (32 - 95) | 0.0 | 14133 ± 7647 | 3360 - 27144 | 54 | 2683 ± 1019 | 3082 ± 1451 | 3726 ± 1715 | 4201 ± 1831 | 4562 ± 1933 | | |
| 4 | Fri | 8/22/14 | 19 (12 - 28) | 62 (30 - 92) | 0.0 | 15921 ± 7935 | 5529 - 35508 | 73 | 3368 ± 2664 | 2487 ± 1034 | 3208 ± 1512 | 3699 ± 1766 | 3960 ± 1890 | | Sea salt |
| 5 | Sat | 8/23/14 | 20 (13 - 28) | 58 (28 - 87) | 0.0 | 15923 ± 7644 | 5595 - 32061 | 72 | 3889 ± 752 | 4708 ± 1309 | 5633 ± 1588 | 6296 ± 1800 | 6860 ± 1894 | | |
| 6 | Tue | 8/26/14 | 19 (16 - 23) | 76 (48 - 96) | 1.3 | 13091 ± 4965 | 5758 - 26105 | 92 | 2765 ± 411 | 5135 ± 1908 | 5961 ± 2242 | 6343 ± 2336 | 6678 ± 2522 | Biomass burning | Biomass burning |
| 7 | Wed | 8/27/14 | 15 (10 - 20) | 79 (55 - 93) | 0.1 | 8627 ± 4965 | 1924 - 20416 | 50 | 1103 ± 549 | 1444 ± 954 | 1822 ± 1304 | 2229 ± 1405 | 2507 ± 1592 | NPF | Sea salt |
| 8 | Thu | 8/28/14 | 13 (10 - 18) | 84 (62 - 96) | 0.0 | 11313 ± 6449 | 4920 - 25556 | 42 | 1432 ± 438 | 2103 ± 1181 | 2672 ± 1388 | 3117 ± 1536 | 3538 ± 1702 | NPF | Sea salt |
| 9 | Fri | 8/29/14 | 13 (12 - 16) | 85 (74 - 91) | 0.0 | 9873 ± 3751 | 5170 - 17513 | 37 | 1609 ± 275 | 2201 ± 529 | 2747 ± 723 | 3146 ± 823 | 3444 ± 861 | NPF | |
| 10 | Sat | 8/30/14 | 16 (12 - 22) | 80 (48 - 94) | 0.0 | 12601 ± 8315 | 4486 - 30528 | 53 | 1945 ± 384 | 3144 ± 1142 | 3856 ± 1380 | 4304 ± 1632 | 4734 ± 1743 | NPF | |
| 11 | Sun | 8/31/14 | 18 (14 - 30) | 83 (30 - 96) | 2.2 | 11014 ± 8580 | 2781 - 18794 | 63 | 1573 ± 679 | 2144 ± 998 | 2525 ± 1159 | 2800 ± 1277 | 3036 ± 1366 | NPF | |
| 12 | Mon | 9/1/14 | 19 (15 - 26) | 75 (48 - 95) | 0.1 | 11014 ± 8580 | 2196 - 31674 | 38 | 782 ± 661 | 1315 ± 1011 | 1959 ± 1273 | 2336 ± 1263 | 2757 ± 1510 | NPF | Biomass burning |
| 13 | Tue | 9/2/14 | 19 (14 - 27) | 79 (46 - 96) | 3.6 | 14842 ± 10075 | 2294 - 32152 | 67 | 2138 ± 661 | 3861 ± 1564 | 4718 ± 1922 | 5281 ± 2073 | 5757 ± 2135 | NPF | |
| 14 | Wed | 9/3/14 | 18 (15 - 25) | 80 (57 - 96) | 2.2 | 7713 ± 7303 | 1347 - 26928 | 29 | 609 ± 328 | 727 ± 428 | 981 ± 580 | 1237 ± 701 | 1523 ± 834 | NPF | |
| Mean | - | - | 17 (10 - 30) | 79 (28 - 98) | 0.9 | 11634 ± 3077 | 1347 - 35508 | 55 | 1987 ± 942 | 2610 ± 1264 | 3218 ± 1435 | 3641 ± 1512 | 3996 ± 1572 | | |

RH – relative humidity; PNC – particle number concentration; Dp – peak diameter; NPF – new particle formation; SS – supersaturation; SD – standard deviation.

**Days with NPF events**

* Emissions from local sources (vehicular traffic, industrial activity, etc.) occur daily, primarily during the diurnal period, and such sources were therefore excluded.

**Remote source apportionment**

**Sea salt** | **Biomass burning**

**Table 2.** Mean activation ratio and particle number concentration (at different supersaturation levels) observed in this study and in other studies.

| Reference | | Site | Environment | Details | Activated ratio (CCN/PNC)* | | | | | PNC ($\times 10^4$) cm$^{-3}$ |
|---|---|---|---|---|---|---|---|---|---|---|
| | | | | | 0.20% | 0.40% | 0.60% | 0.80% | 1.0% | |
| This study | Total mean | Sao Paulo | Urban | Entire campaign | 0.19±0.12 | 0.24±0.16 | 0.30±0.19 | 0.35±0.20 | 0.38±0.22 | 1.2±0.77 |
| | Diurnal | | | - | 0.13±0.08 | 0.16±0.09 | 0.19±0.10 | 0.22±0.11 | 0.52±0.21 | 1.6±0.78 |
| | Nocturnal | | | - | 0.24±0.12 | 0.33±0.15 | 0.41±0.18 | 0.47±0.19 | 0.24±0.12 | 0.3±0.69 |
| Almeida et al.(2014) | | Sao Paulo | Urban | Entire campaign | 0.10±0.05 | 0.19±0.09 | 0.23±0.10 | 0.26±0.11 | 0.28±0.12 | 1.3±0.54 |
| Andreae et al.(2004) | | Amazon | Forest | Green ocean | - | - | - | - | 0.68 | 0.05 |
| Leng et al.(2013) | | Shangai | Urban/Coast | Continental - Marine | - | - | - | 0.4 and 0.6 | - | 0.5 - 1.0 |
| Gunthe et al.,(2011) | | Beijing | Urban | Entire camapaign | 0.41±0.21 | 0.54±0.21 | 0.60±0.23 | 0.66±0.23 | - | 1.7±0.4 |
| | | | | Aged pollution | 0.62±0.08 | 0.74±0.08 | 0.80±0.08 | 0.84±0.06 | - | 1.1±0.23 |
| | | | | Fresh city pollution | 0.15±0.12 | 0.25±0.21 | 0.34±0.24 | 0.42±0.26 | - | 1.2±0.37 |
| Burkart et al.(2011) | | Vienna | Urban | SS = 0.5% | - | 0.13 | - | - | - | 0.9 - 5.4 |
| Sihto et al.(2011) | | Finland | Forest | - | - | 0.21 - 0.42 | - | - | - | ~0.2 |
| Yum et al.(2005) | | Korea | Urban/Coast | Continental - Marine | - | - | - | - | 0.43 - 0.74 | 0.4 - 0.8 |
| Furutani et al.(2008) | | California | Urban/Coast | Anthropogenic polluted | - | - | 0.4 - 0.6[e] | - | - | 0.2 - 1.5 |
| | | | | Fresh Ship exaust | - | - | 0.06 - 0.2 | - | - | 1.5 - 3.0 |
| | | Pacific Ocean | Over Ocean | - | - | - | 0.6 - 1.0 | - | - | 0.2 - 0.6 |

\* CCN – cloud condensation nuclei; PNC – particle number concentration; SS – supersaturation.

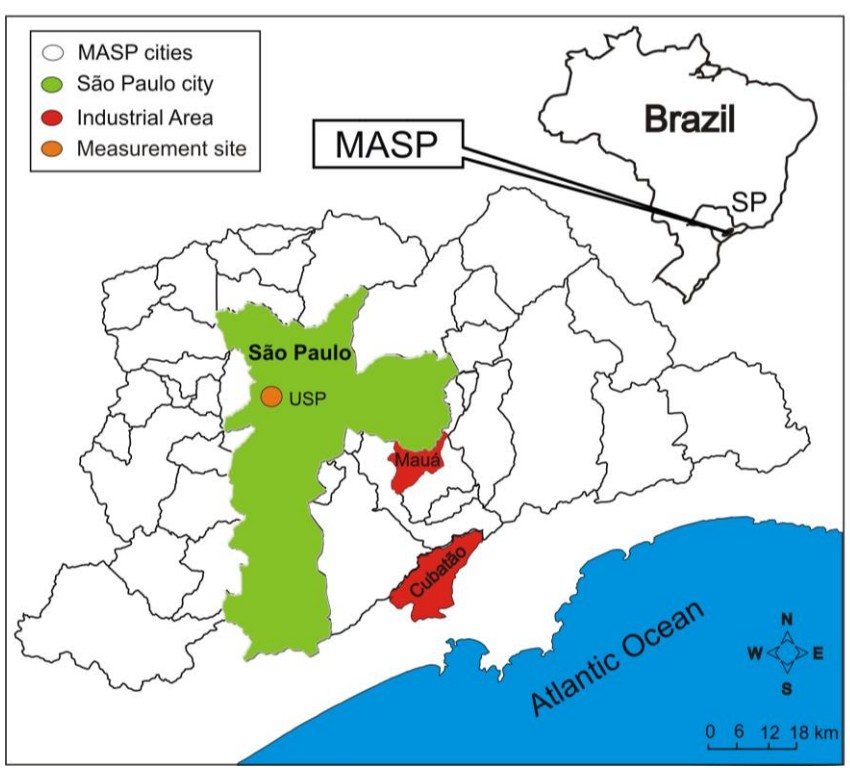

**Figure 1.** Map of Metropolitan Area of São Paulo (MASP). The city of São Paulo proper appears in green. Measurement and sampling sites are located on the campus of the University of São Paulo, represented by an orange circle. The cities of Cubatão and Mauá, both of which comprise large industrial areas appear in red.

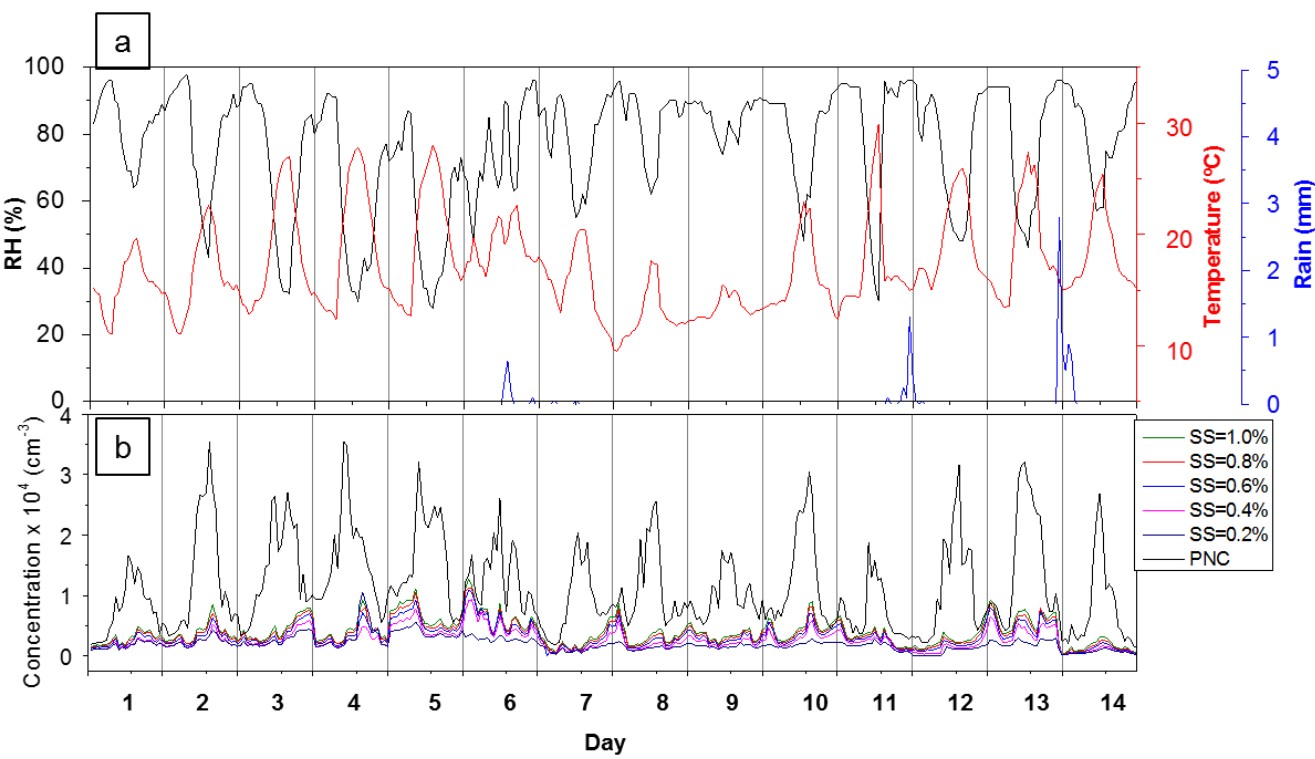

**Figure 2.** Hourly variability in (a) meteorological parameters such as relative humidity (RH), temperature and rainfall, and also for (b) particle number concentration (PNC) and cloud condensation nuclei (CCN) for the studied period. Note that the PNC corresponds to a size distribution of 9–1000 nm, whereas the CCN correspond to a size distribution of 10–2500 nm.

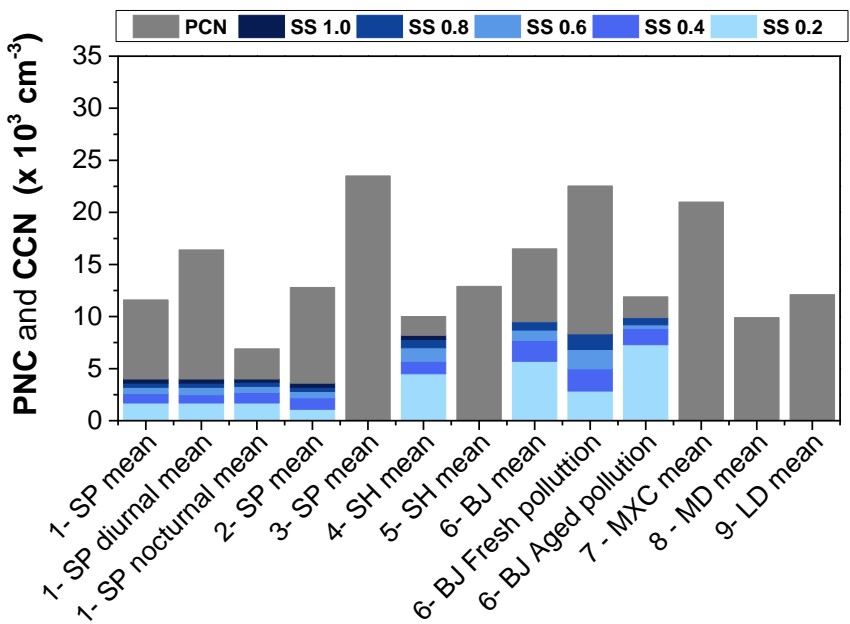

**Figure 3.** Comparison between particle number concentration (PNC) and cloud condensation nuclei (CCN) values obtained in this study and in previous studies. All studies were carried out at urban background monitoring sites, where measurements were made on rooftops of buildings located some kilometres from the downtown areas. In the case of Toronto, the measurements were carried in the downtown area of the city. Detailed information is available on Supplementary information in Table S1.

SP – São Paulo; SH – Shangai; BJ – Beijing; MXC – Mexico City; MD – Madrid; LD – London; 1 –the present study; 2 –Almeida et al. (2014); 3 – Backman et al. (2012); 4 –Leng et al. (2013); 5 – Peng et al. (2014); 6 –Gunthe et al. (2011); 7 - Kalafut-Pettibone et al. (2011); 8 – Gómez-Moreno et al. (2011); 9 –Reche et al. (2011).

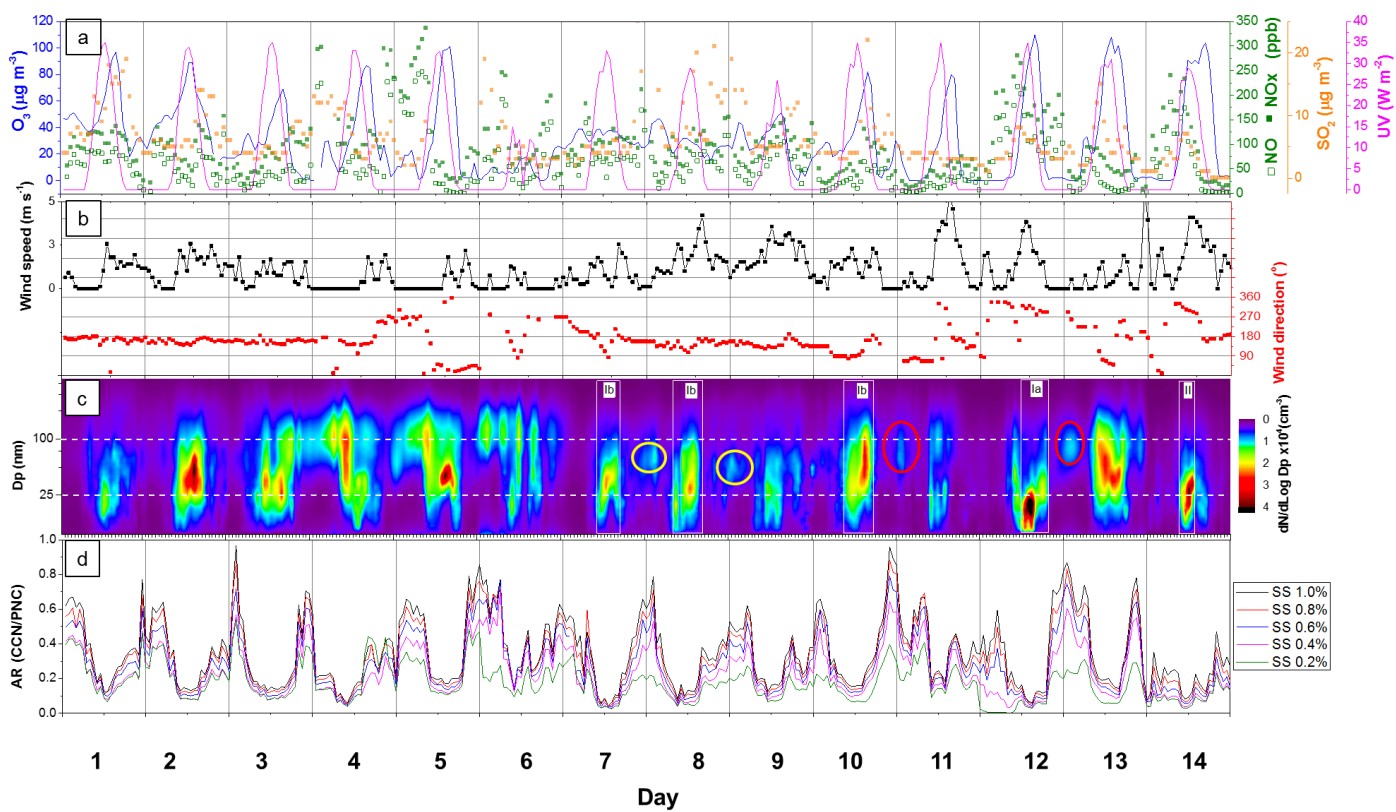

5 **Figure 4.** Hourly variability for (a) UV radiation, together with concentrations of $SO_2$, $NO_x$, $NO_2$ and $O_3$; (b) wind direction and speed; (c) particle number distribution (yellow and red circles indicate conditions of low- and high-$O_3$ concentrations, respectively); and (d) activation ratio (AR), representing the cloud condensation nuclei/particle number concentration (CCN/PNC) ratio. UV radiation data, as well as the concentrations of $SO_2$, $O_3$, $NO_x$ and $NO_2$, were provided by the São Paulo State Environmental Protection Agency.

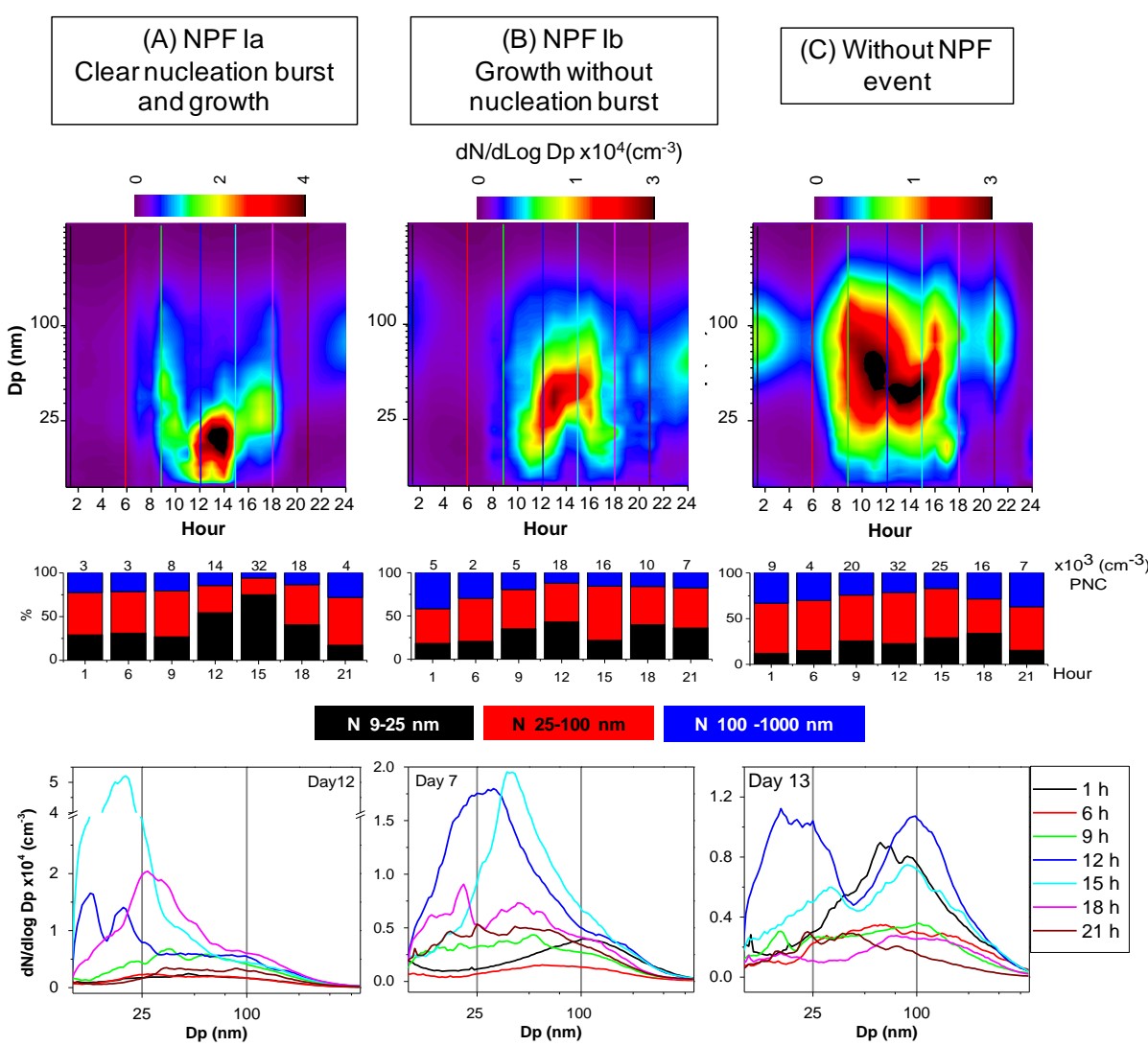

**Figure 5.** Particle number distribution (PND) contour plots for days with (a) new particle formation (NPF) events categorized as class Ia; (b) classIb NPF events; and (c) no NPF events. The hourly bars show the proportions of particle number concentrations (PNCs) in three modes (nucleation, Aitken and accumulation) and hourly line graphs show PND distribution. Selected hours were marked on the PND contour plots in order to compare them with the hourly PND line graphs.

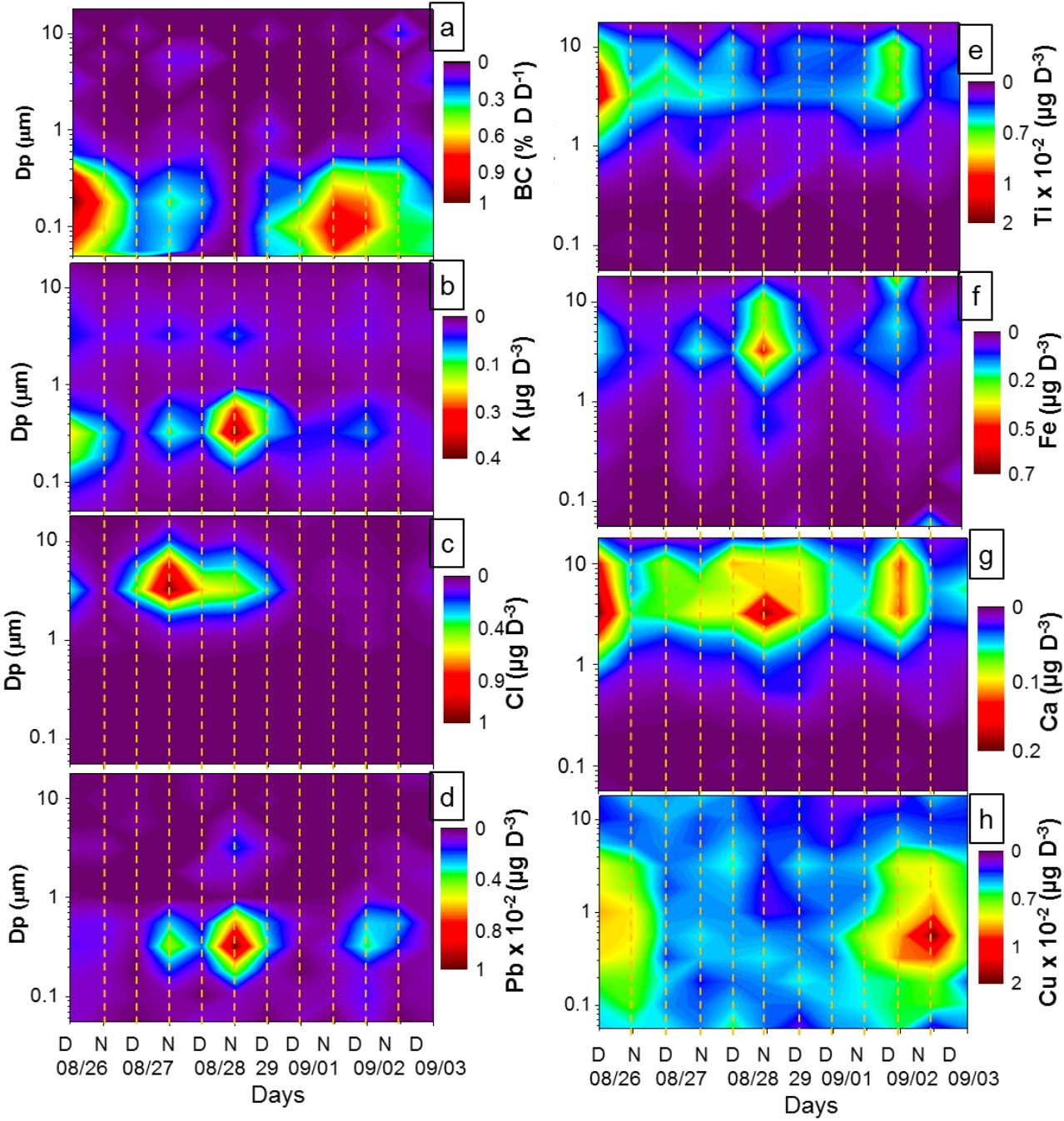

**Figure 6.** Mass size distribution of elemental concentrations of aerosol samples collected by Micro-orifice Uniform Deposit Impactor (a–h) during the diurnal period (0700–1900h) and nocturnal period (1900–0700 h), represented by D and N, respectively. All sampling days were weekdays.

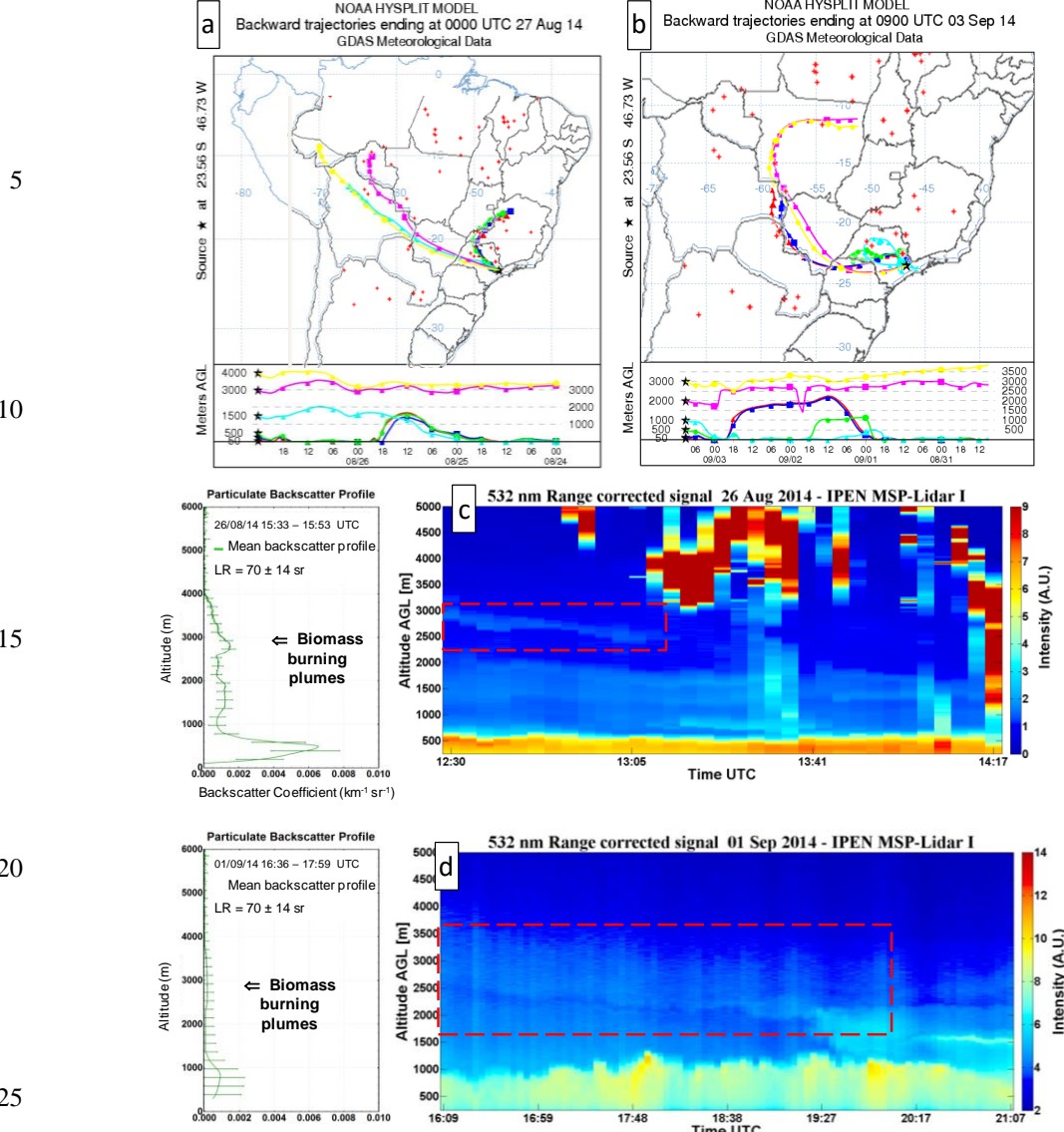

**Figure 7.** Trajectory and lidar analyses for days with biomass-burning events. (a,b) Hybrid Single Particle Lagrangian Integrated Trajectory (HYSPLIT) trajectory models and biomass-burning events for days 6, 7, 12, 13 and 14. (c,d) The particulate backscatter profile and lidar analyses for days 6 and 12, when the prevailing wind was from the north or north-east, where most biomass burning events occur. The biomass burning plumes were detected by lidar and Geostationary Operational Environmental Satellite.

NOAA – (United States) National Oceanic and Atmospheric Administration; GDAS – Global Data Assimilation System; AGL – above ground level; IPEN – *Instituto de Pesquisas Energéticas e Nucleares* ([Brazilian] Nuclear and Energy Research Institute); MSP-Lidar I – Lidar I of the Municipality of São Paulo; LR – lidar ratio.

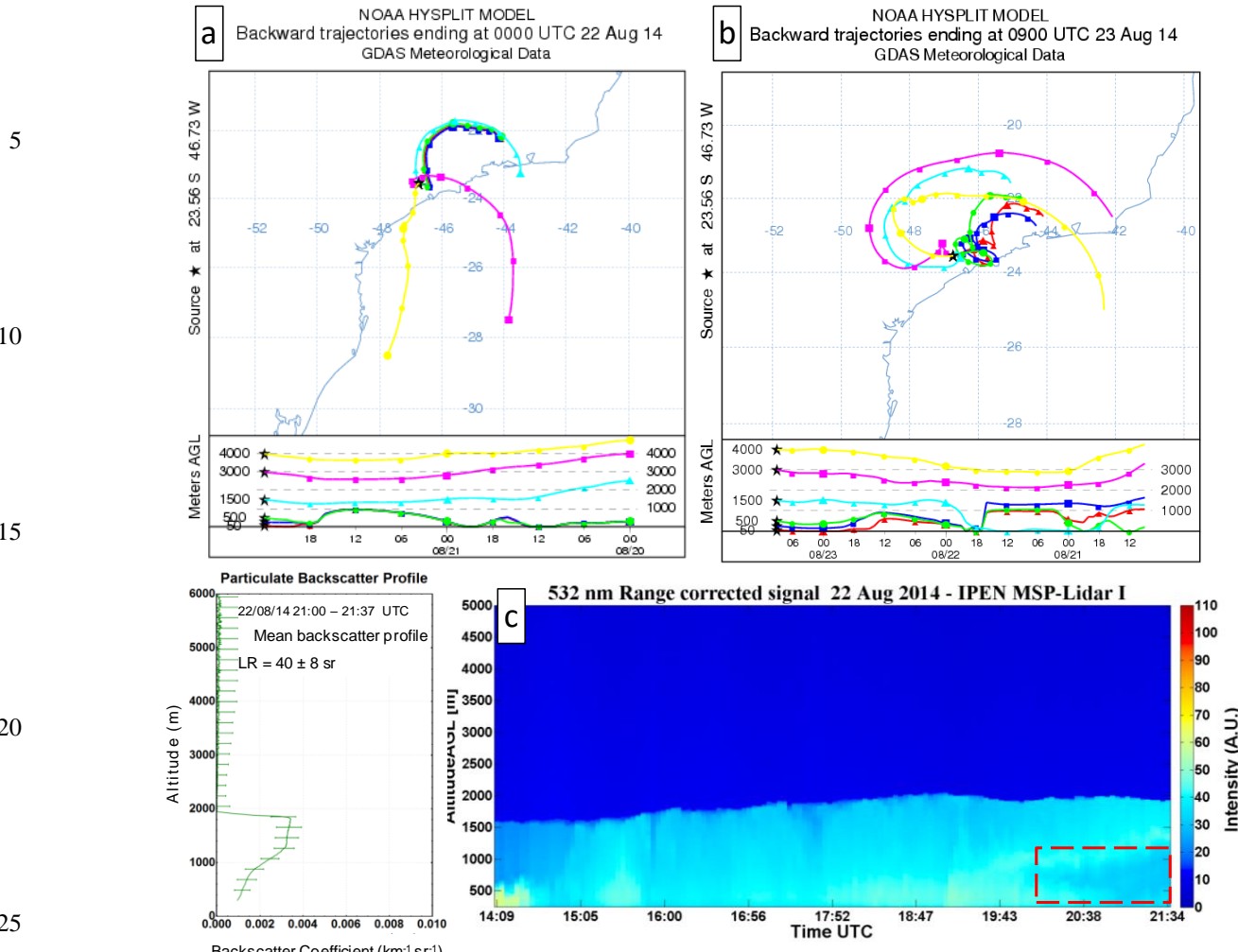

**Figure 8.** Trajectory and lidar analyses for days with sea-salt apportionment. (a,b) Hybrid Single Particle Lagrangian Integrated Trajectory (HYSPLIT) trajectory models for days 3, 4 and 5. (c) Particulate backscatter profile and lidar analyses for days 3 and 4, when the prevailing wind was from the south, south-east or south-west (i.e. from the Atlantic Ocean and coastal regions). Sea-salt plumes were detected by lidar analysis on day 4, although higher confidence interval concentrations were observed on days 7 and 8.

NOAA – (United States) National Oceanic and Atmospheric Administration; GDAS – Global Data Assimilation System; AGL – above ground level; IPEN – *Instituto de Pesquisas Energéticas e Nucleares* ([Brazilian] Nuclear and Energy Research Institute); MSP-Lidar I – Lidar I of the Municipality of São Paulo; LR – lidar ratio.

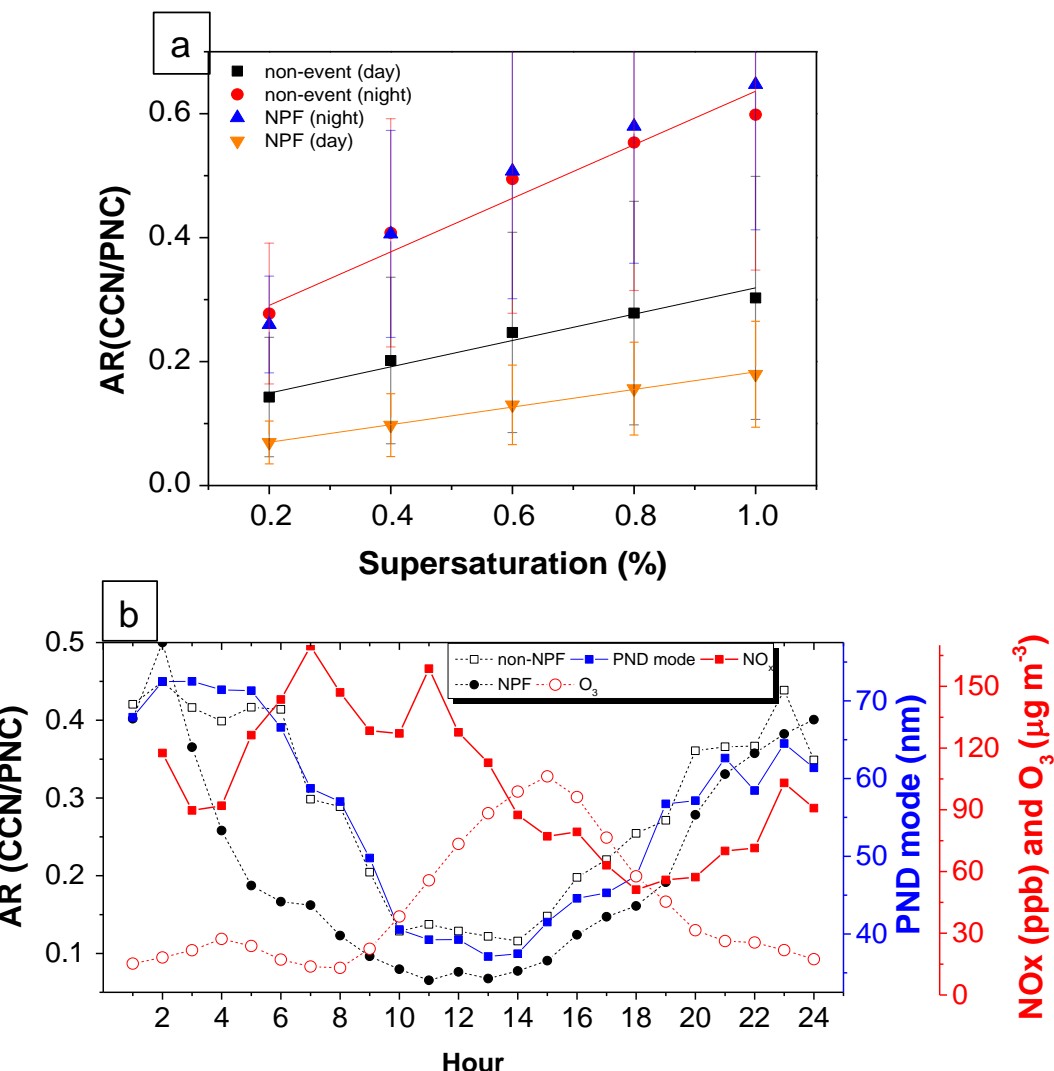

**Figure 9.** (a) Mean activation ratio (AR) for the diurnal and nocturnal periods of days with and without new particle formation (NPF) events, by level of supersaturation. (b) Mean hourly AR (at a supersaturation of 0.4%) on days with and without NPF events, plotted together with particle size mode, nitrogen oxides ($NO_x$) and $O_3$ in order to evaluate the effect of vehicle emissions on particle size and AR. The AR values for nocturnal and diurnal showed clear differences, whereas days with NPF events presented slightly lower values than did days without. Two processes contribute to particle size and AR decrease, namely primary aerosol from vehicular traffic ($NO_x$) and secondary NPF formed in the afternoon.

PND – particle number distribution.

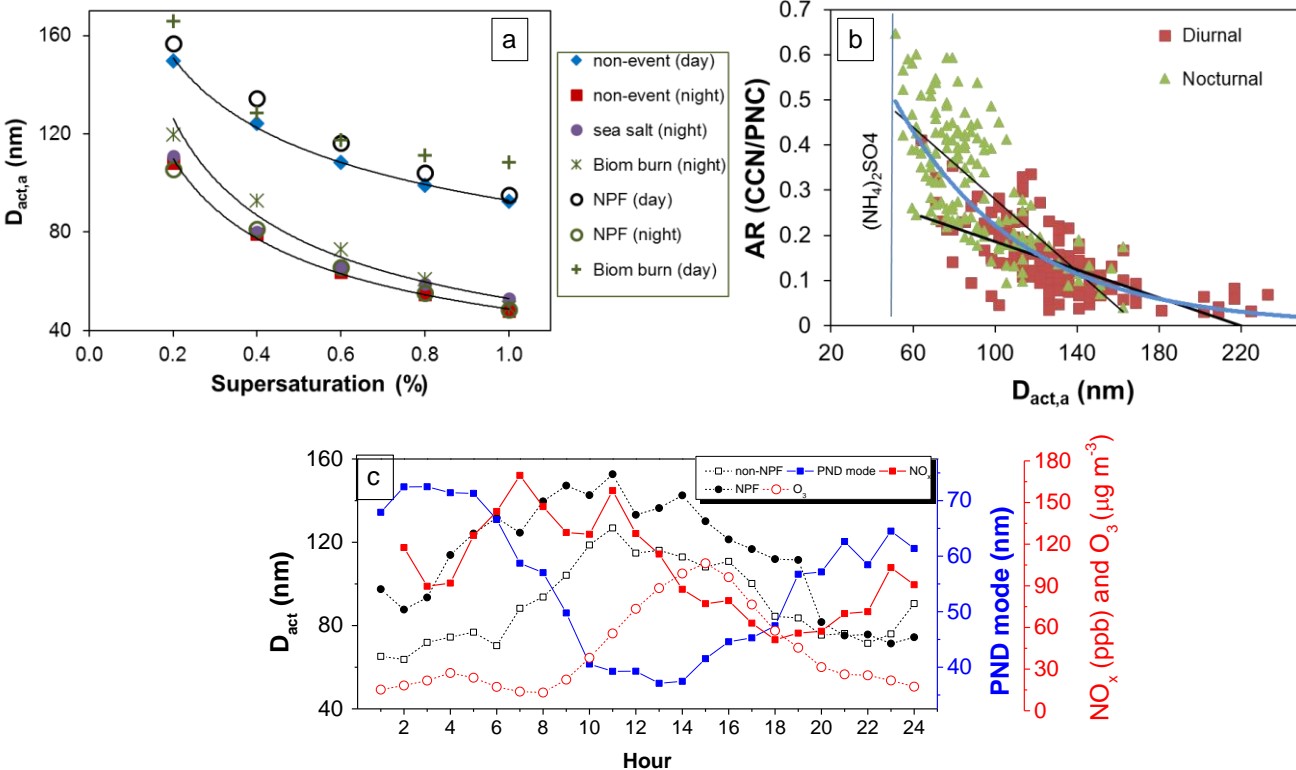

**Figure 10.** (a) Mean apparent activation diameter ($D_{act}$) at various supersaturation levels, showing that it was highest at the lower levels of supersaturation. (b) Activation ratio (AR), in relation to the $D_{act}$. (c) Time series of $D_{act}$ on days with and without new particle formation (NPF) events, plotted together with particle size mode, $NO_x$ and $O_3$. The $D_{act}$ values were calculated for supersaturation of 0.4%.

Biom burn – biomass burning; PND – particle number distribution.

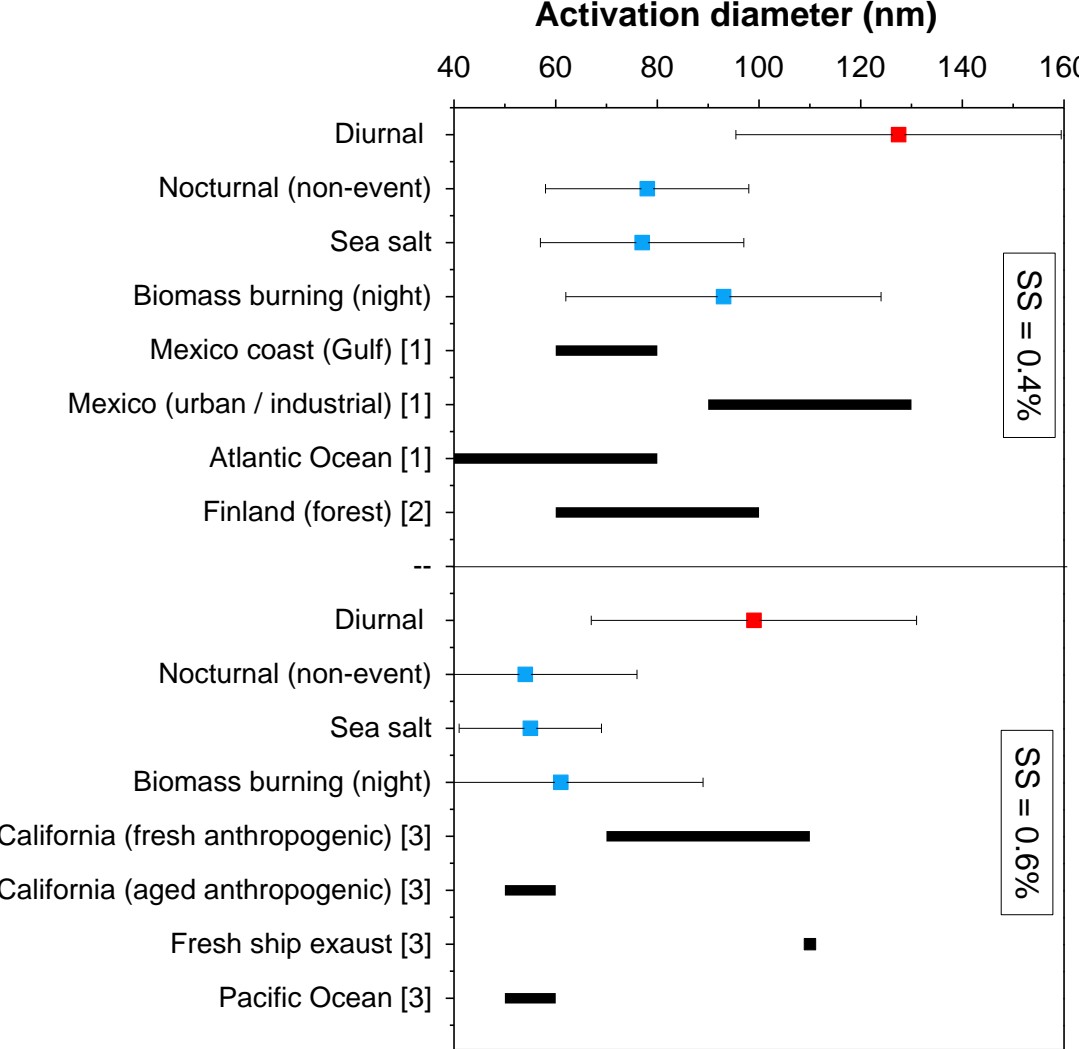

**Figure 11.** Comparison of apparent activation diameter ($D_{act}$) values obtained in this and previous studies, conducted in urban, coastal and forest environments. Mean values for air masses determined to have been influenced by sea-salt transport and biomass burning plumes were calculated for nocturnal periods.

SS supersaturation; [1] Quinn et al.(2008); [2] Sihto et al.(2011); [3] Furutani et al.(2008).

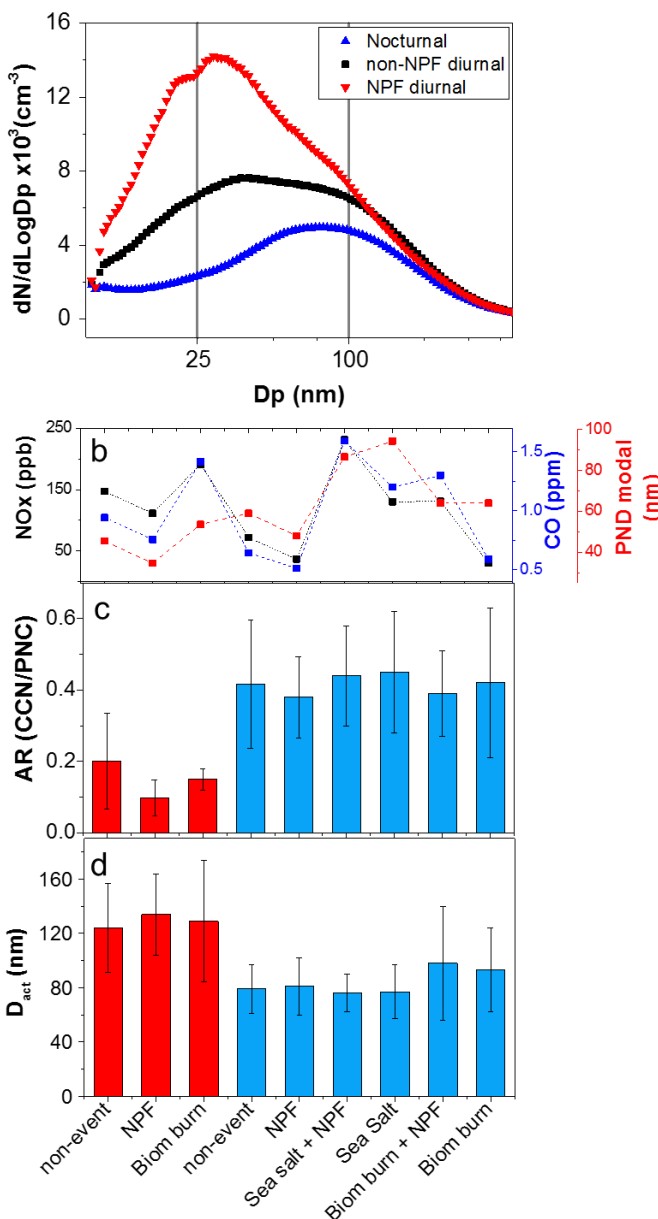

**Figure 12.** (a) Mean diurnal and nocturnal particle size. (b) Mean diurnal and nocturnal concentrations of $NO_x$, CO and PND mode (c) Mean diurnal and nocturnal activation ratio (AR), during new particle formation (NPF), remote source events and non-event periods. (d) Mean diurnal and nocturnal apparent activation diameter ($D_{act}$) during NPF, remote source events and non-event periods. The AR and $D_{act}$ were calculated for a supersaturation of 0.4%.

5    Biom burn – biomass burning; PND – particle number distribution.