# Peer review of "Effect of vehicular traffic, remote sources and new particle formation on the activation properties of cloud condensation nuclei in the megacity of São Paulo, Brazil"

_Atmospheric Chemistry and Physics, 2016_

## Referee Comment (RC1) · Anonymous Referee #1 · 11 Jun 2016

The article deals with the effect of local, remote sources and NPF event on the CCN properties on the basis of observation in a megacity site. Size-resolved aerosol physical and chemical properties were measured, with the assistance of LIDAR and HYS-PLIT, to infer the possible aerosol sources during the campaign. CCN activated ratio and activation diameter were then connected to the effets of those sources. The data could potentially contribute to our understanding of NPF and CCN in the polluted urban atmosphere. I recommend to consider the publication of this manuscript after the following concerns/comments have been addressed:
1. A major conclusion of the work is "local traffic emissions showed higher influence on activation parameters than remote sources". However, from Fig 8 to 11, I can not see any data/information related to traffic source. From Fig 11, I will even conclude that remote sources (seasalt/biomass burning) had more significant influence on AR and Dact. These lead to a more serious problem of this work: the influencing sources of CCN were not clearly classified and many terms were used arbitrarily: diurnal, nocturnal, sea salt, biomass burning, NPF, non-NPF and traffic, the scopes of which overlap with each other. For example, in fig 11 and 10, why are diurnal/nocturnal and seasalt/biomburn put in the same figure? I think "diurnal" aerosols also received contributions from seasalt and biomass burning. Does "diurnal" here include both NPF and non-NPF? In Fig 11a, Fig 8b: "average" of what?

I strongly suggest the authors to state clearly which period (day or night) was dominated by which source. This could be possibly added in Table 1, since there are only 14 observation days. Then CCN properties should be calculated again according to the classification of influencing sources.

2. The current source apportionment is not persuasive and could be a weakness of this work. Three sources were identified: biomass burning, seasalt, and vehicle emission. Industrial source was mentioned a few times in the manuscript, but was omitted in the data analysis. Then the "diurnal" is equivalent, by the authors, to local vehicular emission. It is not shown in the text what elements were analyzed with X-ray Fluorescence. Will these elements together with BC allow a more precise source apportionment?

3. Some judgments made in the abstract are not well supported and probably biased. "weak effects of sea-salt and biomass burning aerosols could be observed on activation parameters as sea-salt showed a positive feedback" "vehicular traffic presented most negative effect on CCN activation." "NPF events showed a negative feedback to CCN activation

First, it was not defined in the text what is negative feedback or positive feedback?
compared with what? second, from the Figures I would say sea salt has significant feedback on CCN parameters, but why is the effect claimed to be "weak"? third, it is generally accepted that NPF will eventually has positive contribution to CCN number. it is thus misleading to say NPF showed negative feedback to CCN activation.

4. Activation Ratio is discussed in this work and shown to have a negative "feedback". But what about CCN number increase/enhancement due to a NPF event? how is it when compared to non-NPF or local/remote episodes?

Other specific comments include: 1. There are many very short paragraphs in the text (like in Section 3.1 and 3.3). They should be reorganized.

2. Some terminology may not be used correctly: "Diurnal" or "daytime"? Page 2 line 26: hygroscope compounds Page 4 line 18: alcohol vapour Page 10 line 2: "frequency of hourly PNCs in three modes", "frequency" or "percentage"?

3. Page 4 line 25-29: is the correction factor 1.3 or 1.15?

4. Some literature citations should be moved to Introduction section, like page 8 line 24-28.

5. Page 15 line 20: "It can be observed that nocturnal samples are most related with water soluble species as (NH4)2SO4, SOA, NO3- and marine air than the diurnal aerosol."——— This is obviously not your observation.

6. There are many other typo- or grammar mistakes in the manuscript.

**ACPD**

---

## Referee Comment (RC2) · Anonymous Referee #2 · 6 Jul 2016

This manuscript contains a potentially interesting data set. The methods used in the paper appear scientifically sound, but the analysis presented in the text has serious deficiencies. Therefore, major revisions, as outlined below, are needed before I can recommend accepting this paper for publication.

Section 2.

I think that the methodology section is relatively well described. The only thing I am a bit skeptical is the discussion about the correction needed for DMPS measurements (lines 22-28 on page 4). Applying a correction factor appears justified due to poten-

tial undercounting of particles. However, the fact that the system does not measure particles larger than 450 nm in diameter is expected to have a negligible effect on this phenomenon (because the fraction of particle number at those size is very small). The authors might consider modifying the text a bit.

Section 3.1.

The comparison of PNC and CCN concentrations to other studies should be made scientifically, not just reporting whether the concentrations observed in other studies had been higher or lower. I recommend that the numerical values of these concentrations, along with those obtained in other studies, will be collected in a Table. There no sense of giving all these numbers in text, rather the text should concentrated on analyzing the differences between this and other studies, and the meaning of these differences.

I understand that the authors compare their PNC data to the earlier Sao Paulo data, but I do not understand the comparison to the Vienna data. Why Vienna and no other urban sites? Also, a reference to Vienna data is missing. I would like to see more urban sites in this PNC comparison.

The comparison needs some logic. There are apparently urban regions of different pollution levels. Is there any systematic pattern between the level of pollution and PNC or CCN concentration? There is enough information in the literature, the authors simply need to have a look at that.

Section 3.2.

This section has several serious problems that need to be fixed.

NO3 radiacals are active during night time only, so it has very little to do with photo-chemistry.

The discussion about SOA formation and its connection with NPF is both outdated and partly erroneous, so should be entirely rewritten in light of more recent literature. SOA formation refers to the secondary production of organic particulate matter, while only

a small fraction of SOA participates in NPF in any way (the least volatile of the gas-phase products). Furthermore, SOA formation itself is not dependent on NPF, since the aerosol volume of surface area needed for SOA formation is almost always dominated by particle larger than those in the nucleation mode. As a result, I see no justification for statements like that in lines 13-14 on page 8, or that in lines 18-19 on page 9.

This discussion about atmospherically-relevant nucleation mechanisms (lines 24-28 on page 8) is seriously outdated.

If mentioning banana and apple –type NPF events, they should be defined somewhere.

Section 3.3

The purpose of this section remains unclear after reading it. The authors discuss connections between a number of tracers and source types, but I have a hard time to catch where all this information is used for in the rest of this paper. I recommend shortening the discussion and summarizing the main findings relevant to the rest of this work in the last paragraph.

The sentence in lines 32-34 on page 10 does not make any sense.

Section 3.4

The third paragraph (lines 18-22 on page 13) discusses AR values related many different environments, yet only two studies have been cited. The sources of all the information referred to here should be explicitly given.

The sentences in lines 23-24 on page 13 are very unclear. . . .increase of AR over SS? What has a diurnal period to do with a slope?

A statement like the one given in lines 3-4 on page 14 need a reference.

Lines 9-18 on page 14: The authors refer to studies mentioned in the introduction without specifying them. This is not a good scientific practice of citing other studies.

The sentence in lines 21-23 starts and ends with a different reference. It remains unclear which information refers to which of these two references.

The sentence in lines 33-34 on page 14 does not make any sense. Furthermore, a citation is missing.

The paragraph in lines 13-22 on page 15 is difficult to follow. The last statement needs a reference. Please rewrite this paragraph.

Finally, the text suffers from rather poor language. Without pointing out individual places in text, there are major problems with many individual sentences, and especially with the use of articles and prepositions (sometimes also with the tense.) After revising the scientific contents of the paper, the authors need be make a very thorough language check out of the text

---

## Author Comment (AC1) · 17 Aug 2016

**Response to reviewer comments for the paper "Effect of local and remote sources and new particle formation events on the activation properties of cloud condensation nuclei in the Brazilian megacity of São Paulo".**

We thank the reviewer for valuable suggestions to improve our manuscript. We agree with the comments and made all the suggested modifications to our revised manuscript. Our responses to each of the comments (in black color) are provided below in blue color. We have highlighted the newly added text with a yellow color in the revised manuscript.

1. A major conclusion of the work is "local traffic emissions showed higher influence on activation parameters than remote sources". However, from Fig 8 to 11, I cannot see any data/information related to traffic source. From Fig 11, I will even conclude that remote sources (sea salt /biomass burning) had more significant influence on AR and Dact. These lead to a more serious problem of this work: the influencing sources of CCN were not clearly classified and many terms were used arbitrarily: diurnal, nocturnal, sea salt, biomass burning, NPF, non-NPF and traffic, the scopes of which overlap with each other. For example, in fig 11 and 10, why are diurnal/nocturnal and sea salt/biomburn put in the same figure? I think "diurnal" aerosols also received contributions from sea salt and biomass burning. Does "diurnal" here include both NPF and non-NPF? In Fig 11a, Fig 8b: "average" of what?

I strongly suggest the authors to state clearly which period (day or night) was dominated by which source. This could be possibly added in Table 1, since there are only 14 observation days. Then CCN properties should be calculated again according to the classification of influencing sources.

Response: In fact,we did not show information related to vehicular traffic in the figs. 8 to 11; hourly NOx average concentrations were added in figs. 9b and 1010, while NOx and CO mean in fig. 12b. The NOx and CO were measured by CETESB in a monitoring station at Marginal Tietê main road, which is distant about 4Km from the sampling site. The variability of NOx in this station is mainly associated with vehicular traffic, once fuel combustion of vehicles is the predominant source in this region. As can be seen in figs. 9b and 10c, NOx increase occurred during a first rush hour in the morning, which is clearly related with a decrease of particles diameter mode and AR (fig. 9b), and also with the increase of activation diameter (fig. 10c). Although the NOx decrease after midday is not associated with the decrease of traffic but with photochemical reactions and $O_3$ formation, briefly commented in Section 3.2, which contribute for secondary aerosol production. Therefore, NOx data corroborate the discussion presented in the next paragraph and included on page 15 in lines 13-19, which associated decrease and lower AR values during the day with vehicular traffic and secondary aerosol nucleation mainly.

The variability in NOx and CO concentrations measured at the Marginal Tietê is mainly associated with vehicular traffic, because fuel combustion is the predominant source of such pollutants in the MASP.As can be seen in Fig. 9b and Fig. 10c, an increase in NOx occurred during the morning rush and was clearly related to the decrease inAR and particle diameter mode (Fig. 9b), as well as to the increase in Dact (Fig. 10c). The NOx decrease after midday was not associated with the decrease in traffic volume but with photochemical reactions and secondary formation of pollutants, as previously discussed. We believe that the NOx and O3 variability corroborates our assumption, explaining the decrease in AR values during the diurnal period in the presence of high vehicular traffic volume and secondary aerosol formation.

[Figure]

Figure 9.(b) - Mean hourly AR (at a supersaturation of 0.4%) on days with and without NPF events, plotted together with particle size mode, nitrogen oxides (NO$_x$) and O$_3$ in order to evaluate the effect of vehicle emissions on particle size and AR. The AR values for nocturnal and diurnal showed clear differences, whereas days with NPF events presented slightly lower values than did days without. Two processes contribute to particle size and AR decrease, namely primary aerosol from vehicular traffic (NO$_x$) and secondary NPF formed in the afternoon. PND – particle number distribution.

[Figure]

Figure 10. (c) - Time series of D$_{act}$ on days with and without new particle formation (NPF) events, plotted together with particle size mode, NO$_x$ and O$_3$. The D$_{act}$ values were calculated for supersaturationof 0.4%. Biom burn – biomass burning; PND – particle number distribution.

We agree that sources of CCN was not clearly described, in order to improve the figs. 11 and 12, and show the influence of each remote source contribution, predominant in different periods, was marked in Table 1 the days under remote source and NPF events, based on sections 3.3 and 3.2 respectively. Local sources represented by vehicular emission, industrial and other sources were not showed on the table 1, because this source present daily emission, being a background condition for urban aerosol at SPMA, consequently local sources apportionment were not discriminated in this study.

Table 1. Diurnal data for meteorological parameters(temperature, relative humidity and rain), particle number concentration and cloud condensation nuclei during the period evaluated (19 August to 3 September, 2014). Peak diameter was calculated by size distribution mode. The last column, represent periods with NPF events and remote sources influence on MASP atmosphere.

| Day | | Date | Temperature (°C) mean (range) | RH (%) mean (range) | Rain (mm) | PNC (cm⁻³) mean ± SD | PNC (cm⁻³) (range) | Dp (nm) | \\multicolumn CCN SS = 0.2% | SS = 0.4% | SS = 0.6% | SS = 0.8% | SS = 1.0% | Remote Diurnal | Source/NPF Nocturnal |
|---|---|---|---|---|---|---|---|---|---|---|---|---|---|---|---|
| 1 | Tue | 8/19/14 | 15 (11 - 20) | 83 (64 - 96) | 0.0 | 7654 ± 4582 | 1874 - 16607 | 50 | 1688 ± 569 | 1809 ± 624 | 2270 ± 830 | 2555 ± 965 | 2811 ± 1029 | | |
| 2 | Wed | 8/20/14 | 16 (11 - 23) | 80 (43 - 98) | 0.0 | 13086 ± 8242 | 3335 - 35398 | 45 | 2240 ± 780 | 2385 ± 961 | 2969 ± 1191 | 3431 ± 1371 | 3770 ± 1611 | | |
| 3 | Thu | 8/21/14 | 18 (13 - 27) | 70 (32 - 95) | 0.0 | 14133 ± 7647 | 3360 - 27144 | 54 | 2683 ± 1019 | 3082 ± 1451 | 3726 ± 1715 | 4201 ± 1831 | 4562 ± 1933 | | |
| 4 | Fri | 8/22/14 | 19 (12 - 28) | 62 (30 - 92) | 0.0 | 15921 ± 7935 | 5529 - 35508 | 73 | 3368 ± 2664 | 2487 ± 1034 | 3208 ± 1512 | 3699 ± 1766 | 3960 ± 1890 | | Sea salt (blue) |
| 5 | Sat | 8/23/14 | 20 (13 - 28) | 58 (28 - 87) | 0.0 | 15923 ± 7644 | 5595 - 32061 | 72 | 3889 ± 752 | 4708 ± 1309 | 5633 ± 1588 | 6296 ± 1800 | 6860 ± 1894 | | |
| 6 | Tue | 8/26/14 | 19 (16 - 23) | 76 (48 - 96) | 1.3 | 13091 ± 4965 | 5758 - 26105 | 92 | 2765 ± 411 | 5135 ± 1908 | 5961 ± 2242 | 6343 ± 2336 | 6678 ± 2522 | Biomass burning (red) | |
| 7 | Wed | 8/27/14 | 15 (10 - 20) | 79 (55 - 93) | 0.1 | 8627 ± 4965 | 1924 - 20416 | 50 | 1103 ± 549 | 1444 ± 954 | 1822 ± 1304 | 2229 ± 1405 | 2507 ± 1592 | NPF (orange) | Sea salt (blue) |
| 8 | Thu | 8/28/14 | 13 (10 - 18) | 84 (62 - 96) | 0.0 | 11313 ± 6449 | 4920 - 25556 | 42 | 1432 ± 438 | 2103 ± 1181 | 2672 ± 1388 | 3117 ± 1536 | 3538 ± 1702 | | |
| 9 | Fri | 8/29/14 | 13 (12 - 16) | 85 (74 - 91) | 0.0 | 9873 ± 3751 | 5170 - 17513 | 37 | 1609 ± 275 | 2201 ± 529 | 2747 ± 723 | 3146 ± 823 | 3444 ± 861 | | |
| 10 | Sat | 8/30/14 | 16 (12 - 22) | 80 (48 - 94) | 0.0 | 12601 ± 8315 | 4486 - 30528 | 53 | 1945 ± 384 | 3144 ± 1142 | 3856 ± 1380 | 4304 ± 1632 | 4734 ± 1743 | | NPF (orange) |
| 11 | Sun | 8/31/14 | 18 (14 - 30) | 83 (30 - 96) | 2.2 | 7078 ± 4543 | 2781 - 18794 | 63 | 1573 ± 679 | 2144 ± 998 | 2525 ± 1159 | 2800 ± 1277 | 3036 ± 1366 | | |
| 12 | Mon | 9/1/14 | 19 (15 - 26) | 75 (48 - 95) | 0.1 | 11014 ± 8580 | 2196 - 31674 | 38 | 782 ± 661 | 1315 ± 1011 | 1959 ± 1273 | 2336 ± 1263 | 2757 ± 1510 | NPF (orange) | Biomass burning (red) |
| 13 | Tue | 9/2/14 | 19 (14 - 27) | 79 (46 - 96) | 3.6 | 14842 ± 10075 | 2294 - 32152 | 67 | 2138 ± 661 | 3861 ± 1564 | 4718 ± 1922 | 5281 ± 2073 | 5757 ± 2135 | | |
| 14 | Wed | 9/3/14 | 18 (15 - 25) | 80 (57 - 96) | 2.2 | 7713 ± 7303 | 1347 - 26928 | 29 | 609 ± 328 | 727 ± 428 | 981 ± 580 | 1237 ± 701 | 1523 ± 834 | NPF (orange) | |
| Mean | - | - | 17 (10 - 30) | 79 (28 - 98) | 0.9 | 11634 ± 3077 | 1347 - 35508 | 55 | 1987 ± 942 | 2610 ± 1264 | 3218 ± 1435 | 3641 ± 1512 | 3996 ± 1572 | | |

RH – relative humidity; PNC – particle number concentration; Dp – peak diameter; NPF – new particle formation; SS – supersaturation; SD – standard deviation.
**Days with NPF events** (orange)

* Emissions from local sources (vehicular traffic, industrial activity, etc.) occur daily, primarily during the diurnal period, and such sources were therefore excluded.
**Remote source apportionment** — **Sea salt** (blue) | **Biomass burning** (red)

Based on days with and without the NPF as well as remote source events, CCN activation properties were recalculated, as shown in figs. 12c and d. The CCN activation properties related to event and non-event days were calculated for diurnal and nocturnal periods and represented in red and blue colors respectively. In addition, were separated means of nocturnal periods after NPF and non-NPF days, in order to evaluate the effects of particles formed during NPF after growth and also exclude this possible effect during biomass burning and sea salt events. Another, information is related to vehicular traffic variability, which was represented by NO$_x$ and CO in Fig. 12b. New discussion related to Fig.12 was included on page 4 in lines 6-26 and also showed on the next two paragraphs.

On the basis of our data for diurnal and nocturnal periods with and without NPF and remote source events, we calculated the mean AR and Dact for specific periods (Fig. 12c and Fig. 12d), in order to identifythe influence thatdifferent events have onCCN activation. The CCN activation properties related to event and non-event days were calculated for the diurnal and nocturnal periods. The variability for diurnal and nocturnal periods was taken into account, and the events were calculated for each period. In addition, we calculated the mean AR and Dactvalues fornocturnal periods after NPF and non-NPF days,in order to evaluate the effects of NPF on CCN activation. Another important information is the variability in the volume of vehicular traffic, asestimated from the meanNOx and CO values(Fig. 12b), which was previouslyassociated with decreasedCCN activation. The overestimation of the volume ofvehicular traffic can result in the underestimation of effects of remote source events onCCN activation.

As can be seen in Fig.12c, diurnal periods with NPF showed lower AR values than did biomassburning and non-event days. However, AR values were similar among nights after NPF with those non-event and sea salt influence nights and also biomass burning after NPF events. The nocturnal increase in AR after NPF events was related to particle growth (Fig. 12c). Lower AR for nocturnal periods with sea-salt

influence after NPF can be attributed to smaller particle size, whereas sea-salt events in these nocturnal periods occurred after low-O3 NPF events (on days 7 and 8). As previously discussed, low-O3 NPFevents were characterised by the formation ofparticles that were smaller than those formed during high-O3NPFevents. The effect of another sea-salt event (day 4) onCCN activation might have been underestimated due tohigh traffic volume during thenocturnal period, as indicated by high concentrations of NOx and CO. The AR values were slightly lower during diurnal and nocturnal periods with biomass burning plumes only (i.e. without NPF events) than during those with non-events. In addition, during the nocturnal periods, the biomass-burning events promotedslightly higher Dact values than did NPF and sea salt, which is attributable to lower particle hygroscopicity. Although there was a lack of statistical significance, weobserved atendency forbiomass burningaffected air masses arriving at the MASP to decrease the activation properties.

**Figure 12.** (a) Average diurnal and nocturnal particle size.Average of CO, NOxand modal particle size(b), theactivated ratio (c) and diameter activation (d) for diurnal and nocturnal periods, as well during NPF, biomass burning and sea salt events.The diurnal period is showed in red and nocturnal in blue.Average for event-NPF were calculated for nights after NPF events.

[Figure]

2. The current source apportionment is not persuasive and could be a weakness of this work. Three sources were identified: biomass burning, seasalt, and vehicle emission. Industrial source was mentioned a few times in the manuscript, but was omitted in the data analysis. Then the "diurnal" is equivalent, by the authors, to local vehicular emission. It is not shown in the text what elements were analyzed with X-ray Fluorescence. Will these elements together with BC allow a more precise source apportionment?

Response: Some paragraphs were rephrased in section 3.3 in order to improve discussion, additionally title of this session was changed to **apportionment of biomass burning and sea salt remote sources,** to state more clearly the objectives of this section, related to the identification of biomass burning and sea-salt influence associated with local pollution sources for MASP aerosol.

In the next two paragraphs follow examples of reviewed paragraphs of section 3.3.

A combination of lidar, HYSPLIT trajectory and size-distributed chemical composition(Na, Ca, Ti, K, Cl P, Fe, Mn, Pb, Cu, Zn, S and BC) analyses were used in order to apportion the contribution of sea-salt and biomass burning withinthe air masses arriving at the MASP. Over the past 30 years, variousstudies conducted in the MASP have employed receptor modelling and aerosol chemical composition analysis in order to identify the main sources of atmospheric pollutants (Bouéres and Orsini, 1981; Andrade et al., 1994; Castanho and Artaxo, 2001; Sanchez-Ccoyllo and Andrade, 2002; Sanchez-Ccoyllo et al., 2008; Andrade et al., 2012). Those studies have determinedthat vehicle emissions account for 50–60% of fine particles, thus constitutingthe main source, followed by oil boilers (accounting for 20–40%), road dust (accounting for 10–30%), industrial emissions(accounting for 10–20%), biomass burning (accounting for 10–20%) and construction activities (accounting for ~10%). Some studies carried on in the MASP evaluated the elemental profiles of the principal urban pollution sources, characterising observed elements such as Mn, Pt, Ni, Cu, Pb, Cr and Zn as markers of gasoline and alcohol emissions (Silva et al.,2010). Diesel burning byheavy-duty vehicles is associated mainlywith BC and S. Suspended road dust, characterised inside and outside road tunnels, has been found to becomprise BC, Si, Al, Fe, Ca, Mg, K, Ti and S, which denotes a mixture of soil, pavement abrasion, tire wear, brakewear and vehicular emission (Hetem and Andrade 2016). In a general approach, other studies found BC to be related to biomass burning;Na and Cl to be related to sea-salt contribution; and Fe, Cu, Zn, Cr, Pb and Ni to be related to industrial emissions (Bzdek et al., 2012; Calvo et al., 2013; Taiwo et al., 2014).

Local pollution sources presents acontinuous contribution to atmospheric aerosol, functioning as background toMASP aerosol, emission of which are higher emission during the diurnal period and lowerduring nocturnal period, as demonstrated by the PNC variability (Fig. 2). The objective of this study was not todiscriminate amongthe main local sources but rather to identify periods during which remote sources were detectable, in order to evaluate how CCN activation properties are affected by such sources, in association with local sources, which cannot be excluded during urban aerosol measurements.

**3.** Some judgments made in the abstract are not well supported and probably biased. "weak effects of sea-salt and biomass burning aerosols could be observed on activation parameters as sea-salt showed a positive feedback" "vehicular traffic presented most negative effect on CCN activation." "NPF events showed a negative feedback to CCN activation.

First, it was not defined in the text what is negative feedback or positive feedback?

compared with what? second, from the Figures I would say sea salt has significant feedback on CCN parameters , but why is the effect claimed to be "weak"? third, it is generally accepted that NPF will eventually has positive contribution to CCN number. it is thus misleading to say NPF showed negative feedback to CCN activation.

**Response:** The abstract was rewritten in order to correct some not well supported or biassed information as follow.

Atmospheric aerosol is the primary source of cloud condensation nuclei (CCN). The microphysics and chemical composition of aerosols can affect cloud development and the precipitation process. Among studies conducted in Latin America, only a handful have reported the impact of urban aerosol on CCN activation parameters such as activation ratio (AR) and activation diameter (Dact). With over 20 million inhabitants, the Metropolitan Area of São Paulo (MASP) is the largest megacity in South America. To our knowledge, this is the first study to assess the impact that remote sources and new particle formation (NPF) events have on CCN activation properties in a South American megacity. The measurements were conducted in the MASP between August and September 2014. We measured the CCN within the 0.2–1.0% range of supersaturation, together with particle number concentration (PNC) and particle number distribution (PND), as well as trace-element concentrations and black carbon (BC). NPF events were identified on 35% of the sampling days. Combining TEC and BC data with an aerosol profile from Lidarand HYSPLITmodel analyses allowed us to identify the contribution of sea salt and biomass burning from remote regions on 28% and 21% of the sampling days, respectively. The AR and Dact parameters showed distinct patterns for diurnal and nocturnal periods. For example, CCN activation was lower during the diurnal periods than during the nocturnal periods, a pattern that was found to be associated mainly with local road traffic emissions. A decrease in CCN activation was observed on the NPF event days, mainly due to high concentrations of particles with smaller diameters. We also found that aerosols from sea salt and biomass burning had minor effects on Dact. For example, nights with biomass burning showed slightly higher Dactvalues than did non-event nights. Our results show that particulate matter from local pollution sources, mainly local road traffic emissions, has a greater effect on CCN activation parameters than those from remote sources.

**4. Activation Ratio is discussed in this work and shown to have a negative "feedback". But what about CCN number increase/enhancement due to a NPF event? how is it when compared to non-NPF or local/remote episodes?**

**Response:** In order to account CCN number enhancement during NPF, days were plotted CCN number mean for NPF and non-NPF days in Figure S2, which was included on Supplementary Information. Could be observed the enhancement of CCN number at nights after NPF events (Fig. S2), caused by th growth of high particle number formed during NPF, being this observation supported by previous studies by experimental measurements or modelling (Sihto et al., 2011). These new text was included in the discussions (page 14, lines 17-21) as follow.

In the present study,the AR increased substantially after each NPF event,although it was still lower than that observed on thenon-event days. In addition, weobserved increase in the number of CCN during thenocturnal periods after NPF events, as showed in Supplementary Information (Fig. S3), attributed tothe high rate of growth amongtheparticlesformed during NPF, an observation that is supported by previous studies (Sihto et al., 2011; Yue et al., 2011; Peng et al., 2014).

[Figure]

**Figure S2.** CCN number concentrations during NPF and non-NPF days. CCN averages were calculated for the48h period, being the first day select as NPF or non-NPF, while the second day was not considerate in this classification. During NPF days the CCN number is lower than non-NPF days in agreement with theprevious discussion about AR showed in Fig. 8. In this figure can be seen clearly the enhancement of CCN number for NPF days in comparison with non-NPF followed by similar values on the second day.

5. Other specific comments include:

1. There are many very short paragraphs in the text (like in Section 3.1 and 3.3). They should be reorganized.

2. Some terminology may not be used correctly: "Diurnal" or "daytime"? Page 2 line 26: hygroscope compounds Page 4 line 18: alcohol vapour Page 10 line 2: "frequency of hourly PNCs in three modes", "frequency" or "percentage"?

3. Page 4 line 25-29: is the correction factor 1.3 or 1.15?

4. Some literature citations should be moved to Introduction section, like page 8 line 24-28.

5. Page 15 line 20: "It can be observed that nocturnal samples are most related with water soluble species as (NH4)2SO4, SOA, NO3- and marine air than the diurnal aerosol."——— This is obviously not your observation.

6. There are many other typo- or grammar mistakes in the manuscript.

Response: Other specific comments:

1. The short paragraphs were reorganized improving the text structure.

All the specific comments as well as rewrite or new paragraphs are plotted in yellow crosshatch in the paper revised version.

2. Diurnal and Nocturnal periods are related to events occurred during daytime and night-time respectively. All other indicate corrections were made.

3. The correction factor employed was 1.3, additional test explaining this factor assurance has now been removed to avoid confusion. In addition, the paragraph was rephrased (page 5, lines 20-25) as follow.

Determining PNCs from the DMPS has been  found to result in the undercounting of particles during ambient measurements, mainly due to lower DMA transfer probability or deviation in sampling and sheath flow rates (Almeida et al., 2014). Another deviation is related to the different diameter size range of particles measured by the DMPS (10–450 nm) and CCN (<10 µm), which can lead to overestimation of AR values, as calculated from the CCN/PNC ratio. A correction factor of 1.3 was applied to the entire data set in order to correct for undercounting during the measurement of PNCs and for overestimation of the AR. That factor was obtained by linear fitting of scatter plot data (CCN versus PNC) with AR values>1.

4. As suggested, some text and citations were moved for introduction section

5. As suggested, we have re-written this phrase (page 16, lines 28-31) as follow.

Our observations support the assumption that nocturnal samples typically comprise greater concentrations of water soluble species, such as $(NH4)2SO4$, SOA, $NO3^-$, and of marine air than do diurnal samples. Our findings are also in keeping with those of other studies showing that aged aerosols present high hygroscopicity (Gunthe et al., 2011; Bougiatioti et al., 2011).

6. We have read the text of the manuscript to remove any grammaltical infelicities.

---

## Author Comment (AC2) · 17 Aug 2016

**Response to reviewer's comments for the paper "Effect of local and remote sources and new particle formation events on the activation properties of cloud condensation nuclei in the Brazilian megacity of São Paulo".**

We thank the reviewer for valuable suggestions to improve our manuscript. We agree with the comments and made all the suggested modifications to our revised manuscript. Our responses to each of the reviewer comments (in black color) are provided below in blue color. We have highlighted the newly added text with a green color in the revised manuscript.

**Section 2 -** I think that the methodology section is relatively well described. The only thing I am a bit skeptical is the discussion about the correction needed for DMPS measurements (lines 22-28 on page 4). Applying a correction factor appears justified due to potential undercounting of particles. However, the fact that the system does not measure particles larger than 450 nm in diameter is expected to have a negligible effect on this phenomenon (because the fraction of particle number at those size is very small). The authors might consider modifying the text a bit

**Response:** The text above about DMPS measurements corrections was rephrased in order to clarify the importance of correction related to AR overestimation.

The aerosol size distribution was measured in the 10–450 nm range, particles being scanned in 22 diameter size bins, with a 5-min time resolution. The gas sample and sheath flow rate were 1.0 and 6.0 L min$^{-1}$, respectively. Determining PNCs from the SMPS has been found to result in the undercounting of particles during ambient measurements, mainly due to lower DMA transfer probability or deviation in sampling and sheath flow rates (Almeida et al., 2014). Another deviation is related to the different diameter size range of particles measured by the SMPS (10–450 nm) and CCN (<10 µm), which can lead to overestimation of AR values, as calculated from the CCN/PNC ratio. A correction factor of 1.3 was applied to the entire data set in order to correct for undercounting during the measurement of PNCs and for overestimation of the AR. That factor was obtained by linear fitting of scatter plot data (CCN versus PNC) with AR values>1.

**Section 3.1** The comparison of PNC and CCN concentrations to other studies should be made scientifically, not just reporting whether the concentrations observed in other studies had been higher or lower. I recommend that the numerical values of these concentrations, along with those obtained in other studies, will be collected in a Table. There no sense of giving all these numbers in text, rather the text should concentrated on analyzing the differences between this and other studies, and the meaning of these differences.

I understand that the authors compare their PNC data to the earlier Sao Paulo data, but I do not understand the comparison to the Vienna data. Why Vienna and no otherurban sites? Also, a reference to Vienna data is missing. I would like to see more urban sites in this PNC comparison.

The comparison needs some logic. There are apparently urban regions of different pollution levels. Is there any systematic pattern between the level of pollution and PNC or CCN concentration? There is enough information in the literature, the authors simply need to have a look at that.

**Response:** In order to improve the comparison of PCN and CCN values of this study, were selected recent studies conducted in urban regions. All values were showed in Fig. 2. Additional information about this regions and studies as well PCN and CCN numerical values, were collected in table S1 and included in the Supplementary information. The text about comparisons and discussion are showed in the follow paragraphs, which were included in the revised version of manuscript (page 8, lines 26-32 and page 9, lines 1-25).

[revised manuscript text omitted]

| Country | Sites | year / period | id | PCN ± SD (x 10³ cm³) | CCN (SS%) (x 10³ cm³) | Instrument | vehicle (million) | inhabitant (million) | Populational density (Km²) | Sampling site | Reference |
|---|---|---|---|---|---|---|---|---|---|---|---|
| Brazil | São Paulo | 2014 (Aug/Sep) 14 days | 1.1 CCN and PCN mean
1.2 Diurnal mean
1.3 Nocturnal mean | 11.6 ± 3.1
16.4 ± 7.9
6.9 ± 3.4 | 2(0.2), 2.6(0.4), 3.2(0.6), 3.6(0.8), 4(1.0)
1.7(0.2), 2.5(0.4), 3.2(0.6), 3.6(0.8), 4(1.0)
1.7(0.2), 2.7(0.4), 3.3(0.6), 3.7(0.8), 4(1.0) | DMT CCN-100 (SS 0.2 - 1.0%) DMPS (10- 450 nm) | 7 | 20 | 2552.0 | Rooftop of building (30m above groung) urban area | This study |
|  | São Paulo | 2012 (Oct - 15 days) | 2 CCN and PCN mean | 12.8 ± 5.4 | 1.1(0.2), 2.2(0.5), 2.8(0.7), 3.2(0.9), 3.6(1.1) | DMT CCN-100 (SS 0.2 - 1.0%) DMPS (10- 450 nm) | 7 | 20 | - | Rooftop of building (40m above groung) urban area | Almeida et al., 2014 |
|  | São Paulo | 2010 Oct - 2011 Jan (79 days) | 3 PCN mean | 23.5 | - | DMPS (6 - 800 nm) | 7 | 20 | - | Rooftop of building near urban area | Backman et al., 2012 |
| China | Shangai | 2010 - 2011 (1 year) | 4 CCN and PCN continental air mean | 10 | 4.5(0.2), 5.7(0.4), 7(0.6), 7.8(0.8), 8.2(1.0) | DMT CCN-100 (SS 0.07 - 2%) DMPS (20-800 nm) | 2.2 | 24 | 3800 | Rooftop of building (30m above groung) residential urban area | Leng et al., 2013 |
|  | Shangai | 2010 (Apr/Jun) | 5 PCN mean | 12.9 | - | DMPS (15-600 nm) | 2.2 | 24 | - | Roof six-floor building, urban residential and business areas | Peng et al., 2014 |
|  | Beijing | 2006 (Aug/Sep) | 6.1 CCN and PCN mean
6.2 Fresh city pollution
6.3 Aged regional pollution | 16.5±9.0
22.5 ± 7.3
11.9 ± 2.8 | 5.7(0.26), 7.7(0.46), 8.7(0.66), 9.5(0.86)
2.9(0.26), 5.0(0.46), 6.8(0.66), 8.3(0.86)
7.3(0.26), 8.8(0.46), 9.2(0.66), 9.9(0.86) | DMT CCN-100 (SS 0.07 - 0.86%) DMPS (3-800 nm) | 2.6 | 18.6 | 1300 | Rooftop third-floor building in a suburban area | Gunther et al., 2011 |
| Mexico | Mexico City | 2006 (Mar) | 7 PCN mean | 21 | - | DMPS (15 - 494 nm) | 4 | 20 | 6000 | Residential and light industrial area | Kalafut-Pettibone et al., 2011 |
| Spain | Madrid | 2007-2008 | 8 PCN mean | 9.9 | - | DMPS (15 - 1000 nm) | 4 | 6 | 5325 | Park inside metropolitan region | Gómez-Moreno et al., 2011 |
| UK | London | 2009 | 9 PCN mean | 12.1 ± 5.8 | - | DMPS (7 - 1000 nm) | 2.6 | 13 | 5223 | North Kensington, surrounded by a mainly residential area. | Reche et al., 2011 |

**Section 3.2** This section has several serious problems that need to be fixed.

$NO_3$ radiacals are active during night time only, so it has very little to do with photochemistry.

The discussion about SOA formation and its connection with NPF is both outdated and partly erroneous, so should be entirely rewritten in light of more recent literature. SOA formation refers to the secondary production of organic particulate matter, while only a small fraction of SOA participates in NPF in any way (the least volatile of the gas-phase products). Furthermore, SOA formation itself is not dependent on NPF, since the aerosol volume of surface area needed for SOA formation is almost always dominated by particle larger than those in the nucleation mode. As a result, I see no justification for statements like that in lines 13-14 on page 8, or that in lines 18-19 on page 9.

This discussion about atmospherically-relevant nucleation mechanisms (lines 24-28 on page 8) is seriously outdated.

If mentioning banana and apple –type NPF events, they should be defined somewhere.

Response: The citation of $NO_3$ radical participation on photochemistry reactions was excluded of paragraph. This one was moved to introduction section, line 10 on page 3, as suggested by another reviewer.

Literature review and text about SOA, NPF and nucleation mechanisms were rewrite and references were updated as follow in the next paragraphs, this new text was included in the reviewed manuscript in lines 10 – 34 on page 3 in the section 1 (introduction).

The cited statements were removed from reviewed version of the manuscript.

[revised manuscript text omitted]

The mention about banana and apple events was removed of discussion on revised manuscript.

Section 3.3 - The purpose of this section remains unclear after reading it. The authors discuss connections between a number of tracers and source types, but I have a hard time to catch where all this information is used for in the rest of this paper. I recommend shortening the discussion and summarizing the main findings relevant to the rest of this work in the last paragraph.

The sentence in lines 32-34 on page 10 does not make any sense.

Response: The discussions were shortened and a final paragraph was included with the conclusions of this section (lines 28-35, page 13) as follow.

In summary, sea-salt air masses arriving at the MASP were observed during the nocturnal period on three of the days evaluated. During the nocturnal period of days 4 and 7, 8, sea-salt events were observed by Lidar and trace-element concentration analysis, respectively. Throughout the year, sea breezes arrive at the MASP in the afternoon and evening (Oliveira et al., 2002; Freitas et al., 2007). In the present study, plumes generated from biomass burning were detected by lidar on days 6 and 12, being associated with an increase in BC on those specific days. In Brazil, numerous biomass burning events occur every year from July to November, mainly in the central and northern regions of the country. However, many such events, associated with agricultural activities, occur within the state of São Paulo throughout the year (Kumar et al., 2016). All focus fire, as shown in figures 7a and 7b, were identified from Geostationary Operational Environmental Satellite images (GOES).

The sentence in lines 32-34 on page 10 was removed.

**Section 3.4**

3.41 The third paragraph (lines 18-22 on page 13) discusses AR values related many different environments, yet only two studies have been cited. The sources of all the information referred to here should be explicitly given.

3.42 The sentences in lines 23-24 on page 13 are very unclear. . . .increase of AR over SS? What has a diurnal period to do with a slope?

3.43 A statement like the one given in lines 3-4 on page 14 need a reference.

3.44 Lines 9-18 on page 14: The authors refer to studies mentioned in the introduction without specifying them. This is not a good scientific practice of citing other studies.

3.45 The sentence in lines 21-23 starts and ends with a different reference. It remains unclear which information refers to which of these two references.

3.46 The sentence in lines 33-34 on page 14 does not make any sense. Furthermore, a citation is missing.

3.47 The paragraph in lines 13-22 on page 15 is difficult to follow. The last statement needs a reference. Please rewrite this paragraph.

3.48 Finally, the text suffers from rather poor language. Without pointing out individual places in text, there are major problems with many individual sentences, and especially with the use of articles and prepositions (sometimes also with the tense.) After revising the scientific contents of the paper, the authors need be make a very thorough language check out of the text.

Response :

3.41 About the third paragraph, all the sources were included in the reviewed manuscript.

3.42 In fact the sentence was unclear, therefore we rewrite this one (lines 29-32, page14 and lines 1-6, page 15) as follow. The comparison between diurnal and nocturnal AR slope, showed in fig. 8a, was imprecise due the high deviation of average values, consequently this sentence was removed.

During the diurnal period, the mean AR values were similar to those observed in other urban areas, although not to those observed in coastal areas (Leng et al., 2013; Furutani et al., 2008), as indicated by recent studies of fresh urban pollution conducted in the MASP, Vienna and Beijing (Almeida et al., 2014; Burkartet al., 2012; Gunthe et al., 2011). However, the AR reported for Beijing was twice that found for the MASP in the present study, considering entire campaign for both, although the PNC values were similar. In addition, the AR values observed for the SPMA in the present study are comparable to the fresh ship exhaust emissions reported in a study conducted along the coast of California (Furutani et al., 2008). The nocturnal AR values observed for the MASP were similar to those reported in a study conducted in a forest environment (Sihto et al., 2011) and in the coastal environments, although opposite those observed in others urban environments (Table 2). However, the mean nocturnal AR and PNC values were higher for aged pollution in Beijing than for the MASP.

3.43 The reference about the statement in lines 3-4 on page 14 was included.

3.44 All the references about mentioned studies in introduction were included.

3.45 In the case of sentence in lines 21-23 on page 14, the information refer to Frank et al.(2006), thus the other citation was excluded.

3.46 The sentence late in lines 33-34 on page 14 was rewrite and highlighted in yellow in the reviewed manuscript (now in lines 15-18, page 16), as follow. In addition, the reference was included.

The mean $D_{act}$ values for diurnal period with biomass burning and NPF events were similar to those observed for non-event days. The $D_{act}$ values for nocturnal periods after NPF or during sea-salt events were similar to those observed after non-event days, although the $D_{act}$ values were slightly higher for nocturnal periods during which there were biomass burning plumes, mainly when the SS < 0.6%. At high supersaturation values, particles with different chemical composition and therefore hygroscopicity have only a weak effect on CCN activity (Sihto et al., 2011; Zhang et al., 2014).

3.47 The paragraph late in lines 13 – 22 on page 15 was rewrite, as follow. In the reviewed manuscript this paragraph is in lines 27 – 35 on page 16, as follow.

The efficiency of aerosol particles to act as CCN can be estimated on the basis of AR and $D_{act}$ data. The AR is dependent on particle size and chemical composition, whereas $D_{act}$ is dependent on chemical composition only (Furutani et al., 2008). As can be seen in Fig. 10b, the non-linear correlation between AR and $D_{act}$ can be related to different chemical composition and size distribution of aerosol. During the diurnal period, the $D_{act}$ was increased and the AR was decreased, whereas the inverse was true for the nocturnal period. In general, the diurnal period is associated with particles that are less hygroscopic and smaller, mainly emitted by vehicular traffic. However, the decreased $D_{act}$ and increased AR were observed in the nocturnal period, being associated with larger and more hygroscopic particles. Our observations support the assumption that nocturnal samples typically comprise greater concentrations of water soluble species, such as $(NH_4)_2SO_4$, SOA, $NO_3^-$, and of marine air than do diurnal samples. Our findings are also in keeping with those of other studies showing that aged aerosols present high hygroscopicity (Gunthe et al., 2011; Bougiatioti et al., 2011).

3.48 We have read the text of the manuscript to remove any grammaltical infelicities and improve the language.

---

## Author Response (AR2)

**Response to reviewer comments for the paper "Effect of vehicular traffic, remote sources and new particle formation events on the activation properties of cloud condensation nuclei in the Brazilian megacity of São Paulo".**

We thank the reviewer for the valuable suggestions to improve our manuscript. We agree with the comments and have made all the suggested modifications. Below, our responses to each of the comments (in black) are provided (in blue). In the revised manuscript, we have highlighted the newly added text in yellow.

1. Abstract and Conclusion sections: "Our results show that particulate matter from local pollution sources, mainly local road traffic emissions, has a greater effect on CCN activation parameters than those from remote sources."

"In summary, local pollution sources, represented mainly by vehicular traffic emissions, had greater negative effects on activation parameters than did remote sources."

According to the authors, local sources are even not able to be discriminated in this study. So how can they conclude local sources had greater negative effect on CCN activation? Moreover, in figure 9, 10, 11 and 12, there are no AR or Dact data about local sources at all. How can they make any conclusion about local source effect? This is my major concern.

Response: In order to assess the main sources of aerosol pollution in the MASP, we carried out a positive matrix factorization (PMF) analysis, using total concentrations of trace elements and BC and carbon monoxide (CO) average concentrations (Fig. S4), as described below. In addition, some markers of vehicular traffic, such as Cr, Cu, Ti and BC, were plotted against carbon monoxide, measured on the main road Marginal Tietê. As can be seen in Figure S5, discussed below, those markers showed strong correlations with CO, which is associated with vehicular emission in urban areas. Our findings are in line with those of many studies, carried out in the MASP, which identified the vehicular traffic as the major local pollutant source in MASP. Those conclusions are in agreement with the great number of vehicles (7 million) in this area and the higher hourly mean vehicular traffic density during the daytime (8000 automobiles, 1500 motorcycles, 220 trucks and 90 buses, on Marginal Tietê). The follow paragraph has been included on page 12, lines 7-19, in the revised manuscript.

In order to assess the source apportionment of aerosol mass concentration, a PMF analysis was applied to trace-element, BC and carbon monoxide (CO) concentrations. The CO is an important marker associated with vehicular emissions in urban areas, its concentrations being measured at the Marginal Tietê. Our analysis showed that vehicular traffic had a great influence on all days, mainly during the daytime, and vehicular emissions + biomass burning had an influence only on days 6, 12 and 13, whereas sea-salt + industrial emissions had an influence only on days 7 and 8 (Figure S4 in the Supplementary Information).

Various studies have identified vehicular traffic as the main local pollutant source in MASP. The contribution of vehicular traffic is higher during the daytime and lower during the nighttime. Some species, such as Cr, Ti, Cu and BC, are frequently used as markers for vehicular traffic (Andrade et al., 2012). In this study, those compounds showed a strong correlation with CO during daytime. We also observed that BC and Cu correlated strongly with CO during nighttime, as illustrated in Figure S5 (Supplementary Information). In addition, the strong correlation between vehicular traffic marker species and CO confirms that vehicular traffic was the predominant daytime source on all days. The strong

correlation of BC and Cu with CO at nighttime can be related to vehicular emission + biomass-burning events, as demonstrated by the PMF analysis.

[Figure]

**Figure S4.** Source apportionment analysis performed by positive matrix factorization (PMF) model, using elemental, carbon monoxide and BC concentrations. Vehicular traffic showed major contributions in most days during daytime mainly, whereas vehicular + biomass burning had main contributions on days 6, 12 and 13. Sea salt and industrial apportionment was most pronounced during days 7 and 8.

[Figure]

**Figure S5.** Elemental concentrations and BC versus carbon monoxide (CO) variability. The strong correlation between vehicular markers species and CO during daytime confirm the major apportionment of this source to aerosol during daytime mainly.

3. page 17 line 20-22,

"As can be seen in Fig.12c, diurnal periods with NPF showed lower AR values than did biomass burning and non-event days. However, AR values were similar among nights after NPF with those non-event and sea salt influence nights and also biomass burning after NPF events."

---In daytime NPF AR is about 0.05 lower than Biomass burning and non-event. Comparing with this, I would say AR in the night after NPF is also lower (not similar!) than those non-event and sea salt influence nights. Or I would say AR in all nighttime classes were just slightly different.

4. page 17 line 23-25

"Lower AR for nocturnal periods with sea-salt influence after NPF can be attributed to smaller particle size, whereas sea-salt events in these nocturnal periods occurred after low-O3 NPF events (on days 7 and 8)."

This sentence does not make sense at all. AR for nocturnal periods with sea salt + NPF is not low. Instead, it is the second highest among all.

Response: Questions 3 and 4 are addressed in the paragraph on page 16, lines 25-30, in the revised manuscript, as follows:

As can be seen in Figure 12b, days with vehicular traffic + NPF showed lower AR values than did those with vehicular traffic + biomass burning and those with vehicular traffic only, without daytime events. In addition, AR values were slightly different at nighttime. The nighttime increase in AR after NPF events was related to particle growth (Fig. 12b). In relation to $D_{act}$, during nighttime, biomass-burning events promoted slightly higher $D_{act}$ values than did NPF and sea-salt transport, which is attributable to lower particle hygroscopicity. Although there was a lack of statistical significance, we observed a tendency for biomass burning to affect the air masses arriving at the MASP by decreasing the activation properties.

6. I suggest to use daytime and nighttime, which are more commonly used in the literature, to replace diurnal and nocturnal throughout the manuscript.

Response: In all instances, the terms nocturnal and diurnal have been replaced with nighttime and daytime in the revised manuscript.

7. I suggest to shorten section 2.6 and 3.4

Response: These sections have been reviewed and shortened in the revised manuscript.

---

## Author Response (AR3)

**Response to reviewer comments for the paper "Effect of vehicular traffic, remote sources and new particle formation events on the activation properties of cloud condensation nuclei in the Brazilian megacity of São Paulo".**

We thank the reviewer for the valuable suggestions to made our manuscript more persuasive. We agree with the comments and have made all the suggested modifications. Below, our responses to each of the comments (in black) are provided (in blue). In the revised manuscript, we have highlighted the newly added text in yellow.

A PMF analysis was performed to apportion the aerosol sources now. Based on PMF, lidar and HYSPLIT analysis, the authors apportioned the aerosols to vehicular traffic, sea salt, industrial and biomass burning. The text and figures were changed accordingly to tightly link daytime aerosol to vehicular traffic. Simultaneous CCN activation parameter measurements then allowed the author to identify the effect of NPF, vehicular traffic and remote sources.

These revision made the manuscript more persuasive. I suggest to accept the manuscript. But the manuscript still needs minor revisions:

1. The PMF apportionment should be added in the abstract.

   We added in the abstract the text about the use of PMF for source apportionment, as follow:

**Abstract.** Atmospheric aerosol is the primary source of cloud condensation nuclei (CCN). The microphysics and chemical composition of aerosols can affect cloud development and the precipitation process. Among studies conducted in Latin America, only a handful have reported the impact of urban aerosol on CCN activation parameters such as activation ratio (AR) and activation diameter ($D_{act}$). With over 20 million inhabitants, the Metropolitan Area of São Paulo (MASP) is the largest megacity in South America. To our knowledge, this is the first study to assess the impact that remote sources and new particle formation (NPF) events have on CCN activation properties in a South American megacity. The measurements were conducted in the MASP between August and September 2014. We measured the CCN within the 0.2–1.0% range of supersaturation, together with particle number concentration (PNC) and particle number distribution (PND), as well as trace-element concentrations and black carbon (BC). NPF events were identified on 35% of the sampling days. Combining multivariate analysis in the form of positive matrix factorization (PMF) with an aerosol profile from lidar and HYSPLIT model analyses, allowed us to identify the main contribution of vehicular traffic in all days and sea-salt and biomass-burning from remote regions on 28% and 21% of the sampling days, respectively. The AR and $D_{act}$ parameters showed distinct patterns for daytime with intense vehicular traffic and nighttime periods. For example, CCN activation was lower during the daytime than during the nighttime periods, a pattern that was found to be associated mainly with local road traffic emissions. A decrease in CCN activation was observed on the NPF event days, mainly due to high concentrations of particles with smaller diameters. We also found that aerosols from sea-salt + industrial emissions and vehicular emissions + biomass-burning had minor effects on $D_{act}$. For example, nights with biomass-burning + vehicular emissions showed slightly lower CCN activation properties than did sea-salt + industrial and non-event nights. Our results show that

particulate matter from local vehicular emissions during the daytime has a greater effect on CCN activation parameters than does that from remote sources.

2. The format of references should be checked. for example, doi is not added for some references. Abbreviation should replace full journal title.

The format of all references and abbreviation for journal title was checked and also the doi was added in all the references.